# Interferon lambda 4 can directly activate human CD19[+] B cells and CD8[+] T cells

Mairene Coto-Llerena[1], Marco Lepore[2], Julian Spagnuolo[2], Daniela Di Blasi[1,2], Diego Calabrese[1], Aleksei Suslov[1], Glenn Bantug[3], Francois HT Duong[1], Luigi M Terracciano[4], Gennaro De Libero[2,*], Markus H Heim[1,5,*]

Compared with the ubiquitous expression of type I (IFNα and IFNβ) interferon receptors, type III (IFNλ) interferon receptors are mainly expressed in epithelial cells of mucosal barriers of the of the intestine and respiratory tract. Consequently, IFNλs are important for innate pathogen defense in the lung and intestine. IFNλs also determine the outcome of hepatitis C virus (HCV) infections, with IFNλ4 inhibiting spontaneous clearance of HCV. Because viral clearance is dependent on T cells, we explored if IFNλs can directly bind to and regulate human T cells. We found that human B cells and CD8[+] T cells express the IFNλ receptor and respond to IFNλs, including IFNλ4. IFNλs were not inhibitors but weak stimulators of B- and T-cell responses. Furthermore, IFNλ4 showed neither synergistic nor antagonistic effects in co-stimulatory experiments with IFNλ1 or IFNα. Multidimensional flow cytometry of cells from liver biopsies of hepatitis patients from IFNλ4-producers showed accumulation of activated CD8[+] T cells with a central memory-like phenotype. In contrast, CD8[+] T cells with a senescent/exhausted phenotype were more abundant in IFNλ4–non-producers. It remains to be elucidated how IFNλ4 promotes CD8 T-cell responses and inhibits the host immunity to HCV infections.

## Introduction

Hepatitis C virus (HCV) is a parenteral transmitted hepatotropic virus that chronically infects an estimated 71 million persons worldwide (WHO, 2017). In most patients, chronic hepatitis C (CHC) leads to some degree of liver fibrosis and in 15–25% cirrhosis develops after 10–40 yr (Lauer & Walker, 2001). Patients with CHC and cirrhosis are at increased risk for liver failure and for developing hepatocellular carcinoma (El-Serag, 2012). Acute HCV infections are often oligo- or asymptomatic (Santantonio et al, 2008). In 70–80% of infected patients, the virus persists and the infection becomes chronic. Clearance of HCV in the acute phase depends on strong and sustained CD4[+] and CD8[+] T-cell responses against multiple peptides within different HCV proteins (Missale et al, 1996; Diepolder et al, 1997; Cooper et al, 1999; Lechner et al, 2000; Takaki et al, 2000; Thimme et al, 2001, 2002). The most direct evidence for the central role of T cells comes from depletion experiments with experimentally infected chimpanzees. Depletion of CD8[+] T cells before experimental infection of previously protected chimpanzees led to HCV persistence until CD8[+] T-cell response recovered and an HCV-specific CD8[+] T-cell response emerged (Shoukry et al, 2003). Furthermore, depletion of CD4[+] cells in previously protected chimpanzees led to HCV persistence and the emergence of CD8[+] escape variants (Grakoui et al, 2003). Collectively, these findings suggested that CD4[+] T cells promote persistence of protective immunity, whereas virus-specific CD8[+] T cells primarily function as the key effectors.

There is a significant association between certain HLA class I (e.g., HLA-B27) and class II (e.g., DRB1*1101) alleles and spontaneous elimination of the virus (Neumann-Haefelin & Thimme, 2013). However, the strongest predictor for spontaneous clearance is a genetic polymorphism in the IFNλ gene locus (Thomas et al, 2009; Rauch et al, 2010; Tillmann et al, 2010). Initially described as the IL28B (IFNλ3) genotype, it has become clear that the originally identified single nucleotide polymorphism rs12979860 and rs8099917 are surrogate markers for the functional single nucleotide polymorphism rs368234815 located in exon 1 of IFNλ4 (Bibert et al, 2013; Prokunina-Olsson et al, 2013). The ancestral allele (designated the ΔG allele) encodes a fully functional IFNλ4 protein, whereas the mutant TT allele encodes an inactive variant with a premature stop codon (Prokunina-Olsson et al, 2013). The impact of this genetic polymorphism on spontaneous clearance is striking: clearance occurs in 50–60% of patients homozygous for the mutant inactive allele, but in only 10–20% of patients with one or two functional alleles (Thomas et al, 2009; Tillmann et al, 2010; Terczynska-Dyla et al, 2014). The association between low spontaneous clearances of HCV with

[1]Department of Biomedicine, Hepatology, University Hospital and University of Basel, Basel, Switzerland   [2]Department of Biomedicine, Experimental Immunology, University Hospital and University of Basel, Basel, Switzerland   [3]Department of Biomedicine, Immunobiology, University Hospital and University of Basel, Basel, Switzerland   [4]Molecular Pathology Division, Institute of Pathology, University Hospital Basel, Basel, Switzerland   [5]Division of Gastroenterology and Hepatology, Clarunis, University Center for Gastrointestinal and Liver Diseases, Basel, Switzerland

Correspondence: markus.heim@unibas.ch; gennaro.delibero@unibas.ch
*Gennaro De Libero and Markus H Heim share co-senior authorship

the IFNλ4 "producer" genotype is statistically significant, but mechanistically unexplained. Conceptually, the simplest mechanistic model predicts that (1) HCV-infected hepatocytes produce and secrete IFNλ4, and (2) IFNλ4 binds to one or more types of immune cells and inhibits the cellular immune response that is critical for HCV clearance. Presently, both assumptions are not supported by direct evidence. So far, IFNλ4 protein could not be detected in liver biopsies of patients with HCV infections. Nevertheless, there is strong indirect evidence that IFNλ4 is a key driver of innate immune responses in HCV infection (Terczynska-Dyla et al, 2014; Heim et al, 2016). The second assumption is also controversial. IFNλ signals through a receptor composed of the ubiquitously expressed IL10RB chain (shared with the IL-10 receptor) and a unique IFNλ receptor chain (IFNλR1) whose expression is mainly restricted to epithelial cells (Kotenko et al, 2003; Donnelly et al, 2004; Sommereyns et al, 2008; Hamming et al, 2013). There are conflicting reports whether human lymphocytes express IFNλR1 and respond directly to IFNλ (Gallagher et al, 2010; Dickensheets et al, 2013). However, there is increasing evidence that IFNλ has immunomodulatory effects on T cells. During acute lymphocytic choriomeningitis virus (LCMV) infection, IFNλ receptor (IFNλR)–deficient mice had increased expansion of CD4+ and CD8+ T cells and enhanced T-cell responses to LCMV re-challenge (Misumi & Whitmire, 2014). These findings led to the hypothesis that IFNλ inhibits T-cell responses. However, because IFNλR could not be detected on T cells and IFNλR-deficient T cells did not respond better than wild-type T cells when transferred in acutely infected WT mice, the authors concluded that the effects on T cells occurred indirectly as a result of IFNλ effects on other cell types (Misumi & Whitmire, 2014).

There is also evidence that IFNλ regulates humoral immune responses, although controversial data were reported in different experimental approaches. Pretreatment of PBMCs from healthy volunteers with IFNλ3 reduced IgG secretion induced by stimulation with H1N1 Influenza virus (Egli et al, 2014). Another study conducted in a mouse vaccination model indicated that IFNλ3 increased vaccine-specific antibody production and IFNγ release (Morrow et al, 2009). A stimulatory effect of IFNλ on antibody production was confirmed in a more recent investigation (Ye et al, 2019). Of note, IFNλ had no direct effect on B cells in mice but acted indirectly by stimulating the secretion of thymic stromal lymphopoietin (TSLP) by M cells in the upper airways. TSLP, in turn, stimulated dendritic cells and boosted immunoglobulin production (Ye et al, 2019). Yet another study found no difference in antibody production between wild-type and IFNλR-deficient mice (Misumi & Whitmire, 2014). In humans, IFNλ might regulate B cells through a direct interaction (de Groen et al, 2015). IFNλR1 was found to be expressed in naive B cells at a very low level (de Groen et al, 2015). Furthermore, stimulation of naive human B cells with IFNλ1 induced IFN stimulated genes (ISGs) and enhanced TLR7/8-induced Ig production (de Groen et al, 2015).

Here we investigated the capacity of major human circulating immune cell populations to directly respond to IFNλ and explored different functional outcomes in responsive cells. Finally, we used multicolor flow cytometry of liver infiltrating lymphocytes to compare T-cell subpopulations in patients with different IFNλ4 genotypes.

# Results

## Human B cells and CD8+ T cells are responsive to IFNλ

The responsiveness of human immune cells to direct IFNλ stimulation is poorly characterized (Sommereyns et al, 2008). Therefore, we investigated whether circulating cells from healthy individuals were sensitive to IFNλ, by monitoring IFNLR-mediated phosphorylation (i.e., activation) of signal transducer and activator of transcription 1 (STAT1). Treatment with IFNλ1 induced STAT1 phosphorylation in PBMCs purified from three healthy donors, albeit at lower levels than the control stimulation with IFNα (Fig 1A). Longer incubation times or higher doses of IFNλ1 did not increase the level of STAT1 phosphorylation (Fig 1B), suggesting that the overall lower response to IFNλ1 could be due to differential IFNλR expression patterns within individual PBMC populations. We determined the responsiveness to IFNλ1 of distinct cell subsets isolated from PBMCs, including total CD3+ T cells, or sorted CD4+ or CD8+ T cells, sorted CD19+ B cells, CD14+ monocytes, and CD3−/CD16+ NK cells. IFNλ1-induced STAT1 phosphorylation (pY-STAT1) was detected in B cells and CD8+ T cells, whereas it was virtually absent in CD14+ cells and in NK cells (Fig 1C). In CD4+ T cells, the pY-STAT1 signal was already detectable in untreated samples, and did not increase upon IFNλ1 stimulation (Fig 1C), suggesting a lack of responsiveness of this subset. Thus, B cells and CD8+ T cells were the only responders to IFNλ1 among PBMC major subsets, and they were probably responsible for the weak signal observed in total PBMCs after IFNλ1 stimulation. Interestingly, both cell populations also responded to IFNλ4 (Fig 1D).

We next determined the gene expression levels of the two components of the type III IFN receptor, namely IFNLR1 and IL10RB, in purified immune cell populations. Consistent with their sensitivity to IFNλ1 and IFNλ4, B and CD8+ T cells displayed higher expression levels of IFNLR1 and IL10RB genes as than the other cell subsets (Fig 1E), whereas the expression of IL10RB and IFNAR1 was comparable in all the cell populations (Fig 1E).

Taken together, these results indicated that the unique capacity of human B cells and CD8+ T cells to respond to IFNλ stimulation might be related with the IFNLR1 gene expression levels found in these cell populations.

## Neutrophils are responsive to IFNλ

Recently, neutrophils have emerged as significant target of IFNλ (Rivera, 2019) expressing functional IFNLR1 and responding to IFNλ stimulation ex vivo by activation of STAT1 phosphorylation (Blazek et al, 2015). In accordance with these reports, IFNλ1 induced of STAT1 phosphorylation in neutrophils isolated from two healthy donors (Fig S1A). Similar to PBMCs (Fig 1A), IFNλ1-induced STAT1 phosphorylation was weaker than IFNα (Fig S1A). IFNλ1 and IFNλ4 showed a weak stimulatory effect on neutrophil migration on their own and did not interfere with fMLP (N-formyl-l-methionyl-l-leucyl-phenylalanine)–induced neutrophil migration (Fig S1B).

## IFNλ4 does not enhance or interfere with TLR-mediated early B-cell activation

We next investigated the impact of IFNλ stimulation on the functional properties of both B cells and CD8+ T cells. We first focused on

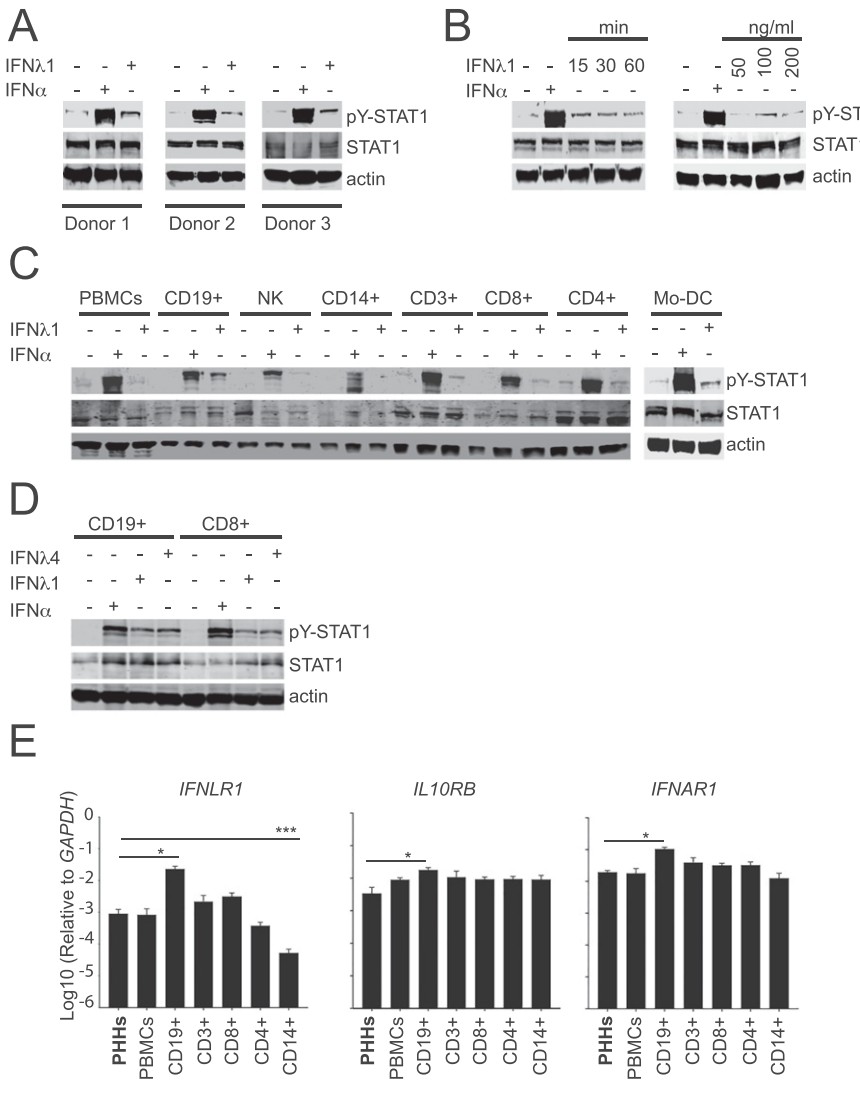

**Figure 1. Identification of IFNλ-responsive cell populations in PBMCs.**
**(A)** PBMCs were stimulated for 15 min with 1,000 IU/ml IFNα or 100 ng/ml of IFNλ1. Western blot detection of phosphorylated STAT1 (pY-STAT1), total STAT1, and actin protein in total cell lysates of PBMCs obtained from three different donors. **(B)** Purified human PBMCs were stimulated with 100 ng/ml of IFNλ1 for 15, 30, and 60 min or 1,000 IU/ml of IFNα for 15 min (left panel) or with 50, 100, and 200 ng/ml of IFNλ1 or 1,000 IU/ml of IFNα for 15 min. Shown are representative blots from two different donors. **(A, C)** Total PBMCs and purified CD19⁺ B cells, CD3⁺ cells, CD8⁺ T cells, CD4⁺ T cells, CD3⁻/CD16⁺ NK cells, and CD14⁺ monocytes were stimulated for 15 min with 1,000 IU/ml IFNα or 100 ng/ml IFNλ1 and analyzed as described in (A). A representative blot from two experiments is shown. **(A, D)** PBMC-derived CD19⁺ B cells and CD8⁺ T cells were stimulated for 15 min with 1,000 IU/ml IFNα, 100 ng/ml IFNλ1, or 100 ng/ml IFNλ4 and analyzed as described in (A). **(E)** qRT-PCR analysis of *IFNLR1*, *IFNAR1*, and *IL10RB* transcripts in total RNA isolated from the indicated PBMC subpopulations and primary human hepatocytes. Transcript levels are expressed as the ΔΔCT relative to *GAPDH*. Results are shown as mean ± SEM; n = 3.

B cells. According to published studies, both IFNα and IFNλ1 enhance B-cell responses to TLR ligands (Bekeredjian-Ding et al, 2005; de Groen et al, 2015). In contrast, the role of IFNλ4 in regulating B-cell activation is not known. Thus, we asked whether IFNλ4 could modulate B-cell functions. In a first series of experiments, purified CD19⁺ B cells were stimulated for 48 h with saturating doses of IFNα, IFNλ1, or IFNλ4 in the absence of TRL ligands and analyzed by flow cytometry for the expression of the early activation marker CD69. IFNα induced a marked and statistically significant increase in the frequency of CD69⁺-activated B cells and of CD69 median fluorescence intensity (Fig 2A). IFNλ1 induced a slight but statistically significant increase, whereas IFNλ4 failed to promote a comparable response.

Of note, IFNλ4 did not alter the frequency of CD69⁺ B cells also when used in combination with IFNα or IFNλ1 (Fig 2B), thus suggesting lack of synergistic or antagonistic effect with IFNα and IFNλ1 on early B-cell stimulation. All three IFNs induced the expression of the ISGs *IFI27*, *Mx1*, *IFIT1*, and *OAS1* (Fig 2C). This finding confirmed that B cells may respond to IFNλs and suggested that the differences between IFNα and IFNλs in regard to B-cell activation are not due to nonspecific differences in

signal transduction. Of note, no antagonism or synergism was observed between all IFNs in regard to ISG induction (Fig 2C).

We next investigated whether IFNλ4 stimulation could impact TLR-mediated activation of B cells. In agreement with previous studies (de Groen et al, 2015), IFNα and IFNλ1 significantly enhanced TLR-mediated early B-cell activation as measured by an increase in both frequency of CD69⁺ B cells and CD69 MFI (Fig 2D). Conversely, IFNλ4 had no effect on TLR9-dependent B-cell stimulation (Fig 2D), nor inhibited the B-cell stimulatory activity of IFNα, IFNλ1, or CpG (ODN2006) (Fig 2E). Collectively, these data suggested that the B cells responsiveness to IFNλ4 did not result in significant functional changes in the early B-cell activation program. Furthermore, IFNλ4 did not interfere with the capacity of IFNα, IFNλ1, or CpG (ODN2006) to stimulate early B-cell activation.

## IFNλ4 is a weak inducer of IL-10 production by B cells

IL-10 is a potent inhibitor of cellular immune responses and is released by several cell types, including activated B cells (Blair et al,

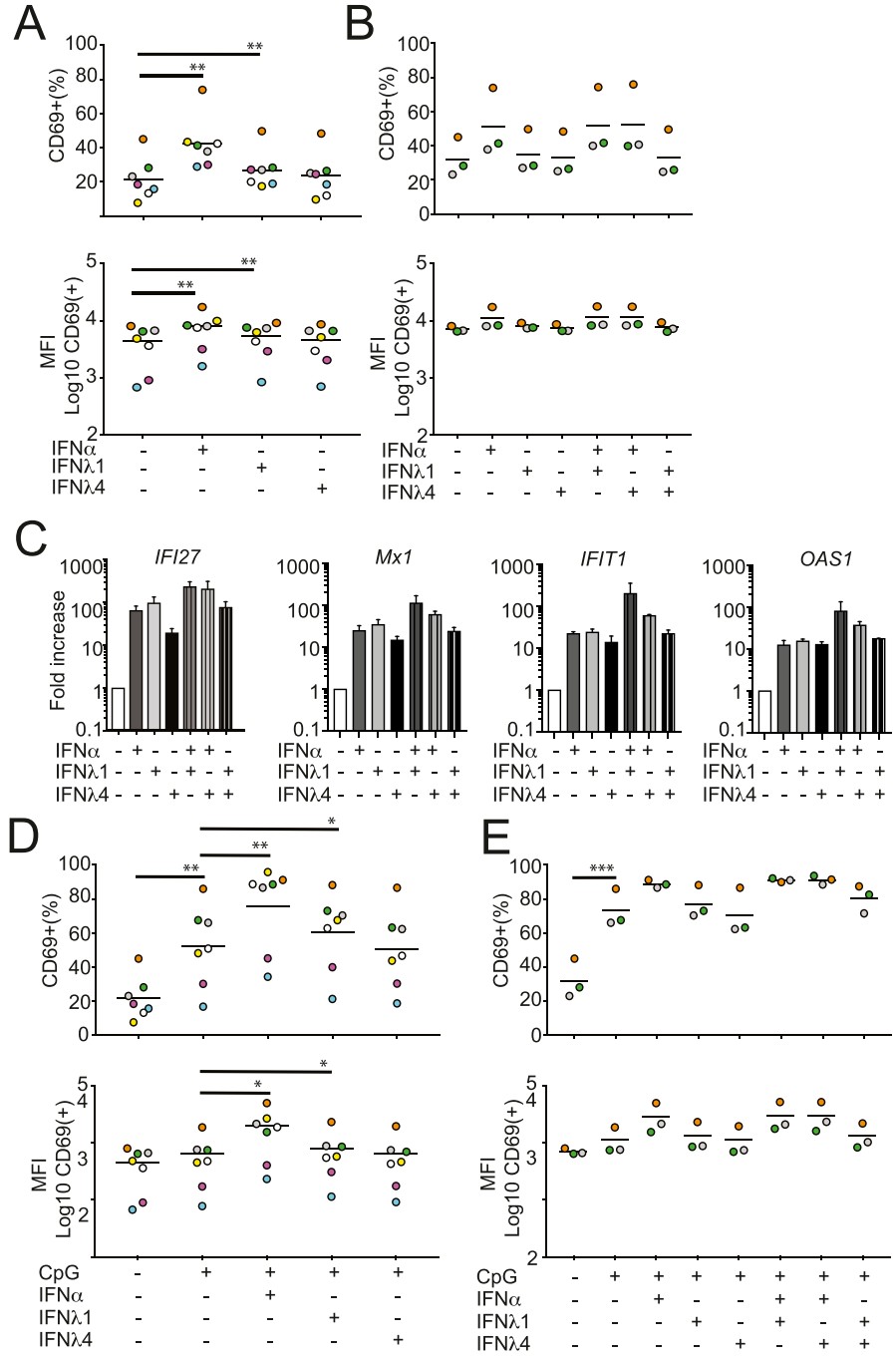

**Figure 2.  IFNλ4 has no co-stimulatory activity on B-cell activation.**
B cells were isolated from PBMCs and stimulated with IFN and/or CpG ODN2006 for 48 h and analyzed for B-cell activation. **(A)** B cells isolated from seven donors were stimulated with IFNα (1,000 IU/ml), IFNλ1 (100 ng/ml) and IFNλ4 (100 ng/ml), fixed, and analyzed by CD69-specific mAbs by FACS. Percentages of CD69[+] cells within the B-cell population (upper panel) or median fluorescence intensity (MFI) of CD69[+] cells within the B-cell population (lower panel) are shown. Significant changes between IFN-treated and control samples are denoted by a thick line. **(A, B)** B cells isolated from three donors were stimulated with IFNα (1,000 IU/ml), IFNλ1 (100 ng/ml), and IFNλ4 (100 ng/ml) alone or in combination and analyzed as described in (A). Significant changes between IFN-treated and control samples as well as combination versus single IFN treatment are denoted by a thick line. **(C)** *IFI27*, *MX1*, *OAS1*, and *IFIT1* transcript levels in B cells stimulated with individual or combinations of IFNα, IFNλ1, and IFNλ4 were analyzed by qRT-PCR of total RNA. Results are shown as ΔΔCT relative to *GAPDH* (mean ± SEM; n = 2). **(D, E)** B cells were stimulated with IFNα (1,000 IU/ml), IFNλ1 (100 ng/ml), and IFNλ4 (100 ng/ml) alone (D) or in combination (E) in the presence or absence of CpG (0.8 μg/ml). **(A)** Cells were analyzed as described in (A). **(A, B, D, E)** Colors depict individual donors and thin horizontal lines indicate the mean; n = 7 for (A) and (D); n = 3 for (B) and (E). *P < 0.05, **P < 0.01, and ***P < 0.001 (paired *t* test).

2010; Das et al, 2012). High levels of IL-10 have been linked with immune-suppression in multiple chronic infections, including HCV (Lund, 2008; Ng & Oldstone, 2014). We hypothesized that IFNλ4 stimulation could promote IL-10 secretion by activated B cells. CD19[+] enriched B cells were stimulated for 48 h with IFNs in the presence or absence of CpG (ODN2006), a TLR9 ligand. Culture supernatants were then analyzed for IL-10 protein levels by ELISA. In the absence of TLR9 triggering, IFNs did not induce IL-10 se-cretion by B cells (Fig 3A). When individually combined with CpG (ODN2006), all IFNs promoted a significant increase in IL-10

production, with IFNλ4 being the weakest stimulator (Fig 3A). In-terestingly, we did not find any synergistic or antagonistic effect of IFNλ4 when used in combination with IFNλ1 or IFNα (Fig 3A). In agreement with previous studies (Banko et al, 2017), the CD19[+]/CD27[+] B cells, which are enriched in the memory B-cell population, were the main source of IL-10 upon stimulation with CpG and IFNλ1 or IFNλ4 (Fig 3B).

Another effect of TLR9 ligation in CD19[+]/CD27[+] memory B cells is induction of IgG release in the absence of B-cell receptor stimu-lation (Bernasconi et al, 2003). This response was not significantly

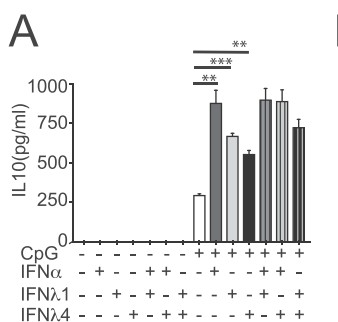
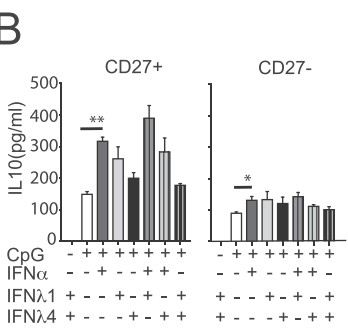
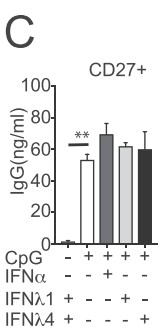

**Figure 3. IFNλ4 induces IL-10 production in B cells.**
**(A)** B cells (n = 4) were stimulated with IFNα (1,000 IU/ml), IFNλ1 (100 ng/ml), and IFNλ4 (100 ng/ml) alone or in combination and in the presence or absence of CpG (2.5 µg/ml) for 48 h. Cell supernatants were collected and released IL-10 was determined by ELISA. Results shown as ± SEM from two independent experiments. Significant changes between IFN-treated and control samples as well as IFN combinations versus single IFN treatment are denoted by a thick line. **(B)** Memory (CD27⁺ CD19⁺) and naïve (CD27⁻ CD19⁺) B cells were stimulated with IFNα (1,000 IU/ml), IFNλ1 (100 ng/ml), and IFNλ4 (100 ng/ml) alone or in combination for 48 h either in the absence or presence of CpG (2.5 µg/ml). IL-10 present in supernatants of TLR9-stimulated and unstimulated cells in the presence of IFN was analyzed by ELISA. Results show mean ± SEM concentration (n = 4). **(C)** Memory B cells were treated with IFNα (1,000 IU/ml), IFNλ1 (100 ng/ml) or IFNλ4 (100 ng/ml) in the presence of CpG (2.5 µg/ml) for 48 h. The presence of IgG in the supernatants of stimulated and unstimulated controls was analyzed by ELISA. Results show the mean ± SEM concentration (n = 4). *$P < 0.05$, **$P < 0.01$, and ***$P < 0.001$ compared with the control condition (paired $t$ test).

increased by IFNα or IFNλ1 (de Groen et al, 2015). We investigated whether IFNλ4 could modulate IgG production by CpG (ODN2006)-stimulated purified memory B cells. IFNλ4, like IFNα or IFNλ1 (de Groen et al, 2015), did not increase nor impair CpG (ODN2006)-induced IgG secretion (Fig 3C). Altogether, these data suggested that in memory B cells IFNλ4 promotes similar but less pronounced functional responses than those induced by other type III and type I IFNs.

## IFNλ4 enhances IFNγ production by TCR stimulated CD8 T cells

Alike B cells, CD8⁺ T cells showed response to IFNλ (Fig 1C). Thus, we asked whether IFNλ could modulate CD8⁺ T-cell functions. Purified CD8⁺ T cells were stimulated for 48 h with saturating doses of IFNλ1 or IFNλ4 in the presence or absence of titrating doses of plastic-bound anti-CD3 mAbs. Secreted IFNγ, a cytokine with potent antiviral activity, including HCV (Slifka & Whitton, 2000), was measured in culture supernatants as readout of response.

IFNλ1 and IFNλ4 did not trigger any IFNγ production by CD8⁺ T cells in the absence of plastic-bound anti-CD3 mAbs (Fig 4A). Instead, they both significantly enhanced IFNγ secretion (2.94 ± 0.71 and 2.85 ± 0.38-fold, respectively) in the presence of suboptimal doses (0.1 µg/ml) of anti-CD3 mAbs (Fig 4A). Such synergistic effect with TCR-mediated stimulation was lost when higher doses of anti-CD3 mAbs were used (Fig 4A). Additional experiments with a fixed suboptimal dose of plastic-bound anti-CD3 mAbs and limiting doses of IFNλ1 or IFNλ4 confirmed a dose-dependent relation between the amount of IFNγ released by CD8⁺ T cells and the concentrations of IFNλ1 or IFNλ4 (Fig 4B). Thus, both IFNλ1 and IFNλ4 enhance secretion of IFNγ by CD8⁺ T cells in conditions of suboptimal TCR stimulation with anti-CD3 mAbs, which more closely resemble physiological antigen stimulation. Next, we investigated whether IFNλ has also synergistic effects on secretion of other cytokines by CD8⁺ T cells (Fig 4C–H). IFNλ1 significantly stimulated IL-10 and IL-1β in anti–CD3-activated cells but had no significant impact on secretion of IL-22, GM-CFS, or TNFα.

CD8⁺ T cells can be classified according to the differential expression of two surface markers (CD45RA and CCR7) in naïve (T_N, CD45RA⁺/CCR7⁺), central memory (T_CM, CD45RA⁻/CCR7⁺), effector memory (T_EM; CD45RA⁻/CCR7⁻), and CD45RA⁺ effector memory (TEMRA; CD45RA⁺/CCR7⁻) cells (Rufer et al, 2003; Sallusto et al, 1999). These subsets display distinct differentiation states, tissue-homing properties, and functional profiles. We therefore investigated

whether they also showed differential responsiveness to IFNλ stimulation. Individual CD8⁺ T-cell subpopulations were purified, stimulated with 1,000 IU/ml IFNα or 100 ng/ml IFNλ1 and analyzed by Western blotting for activation of STAT1 (pY-STAT1). All populations responded to IFNα stimulation (Fig 4I). In contrast, IFNλ1 induced phosphorylation of STAT1 predominantly in TEMRA cells and only to a lesser extent in the T_CM cells. T_N and T_EM did not show a response to IFNλ1 stimulation. As TEMRA cells mount rapid and robust IFNγ responses (Sallusto et al, 1999) and given their unique sensitivity to IFNλ1, these cells are likely to be the major targets for the IFNγ-boosting properties of IFNλ.

Next, we investigated whether IFNλ could synergize with IFNα in enhancing CD8⁺ T-cell responses induced by suboptimal TCR-triggering (Nguyen et al, 2002; Curtsinger et al, 2005; Hervas-Stubbs et al, 2010). Combinations of IFNα, IFNλ1, and IFNλ4 in the absence or presence of suboptimal doses of anti-CD3 mAbs were therefore used. No synergistic or antagonistic effects of IFNs were observed (Fig 4J). Taken together, these results indicated that IFNλs might amplify CD8⁺ T-cell effector functions induced by suboptimal antigenic stimuli, similarly to what has been described for IFNα (Curtsinger et al, 2005). Furthermore, no inhibitory effect of IFNλ4 on IFNγ production induced by IFNα or IFNλ1 was observed.

## Impact of IFNλ4 genotype on intrahepatic T cells in patients with CHC

To investigate whether there were main differences in T cells between IFNλ4 genotype ΔG and TT patients, immunohistochemical analysis was performed on liver tissue of a cohort of CHC patients. We focused on determining the relative frequencies of PD1⁺ T cells in the liver as this marker is expressed on activated T cells and is associated with inhibition of the immune response. Accordingly, we performed CD3/PD1 co-staining on sections of paraffin-embedded liver tissue (Fig 5B and C) obtained from 20 CHC patients of IFNλ4 genotype ΔG and 19 of IFNλ4 genotype TT (Table 1). This analysis revealed that there was no difference in the frequency of CD3⁺/PD1⁺ cells in the liver of ΔG and TT CHC patients (Fig 5A–C), thus excluding differential accumulation of T cells and expression of PD1 as reason of virus persistence in IFNλ4-producer patients.

The functional differences between ΔG and TT patients might be associated with more subtle changes in unique populations of T cells. Therefore, we performed a multicolor Flow cytometry analysis

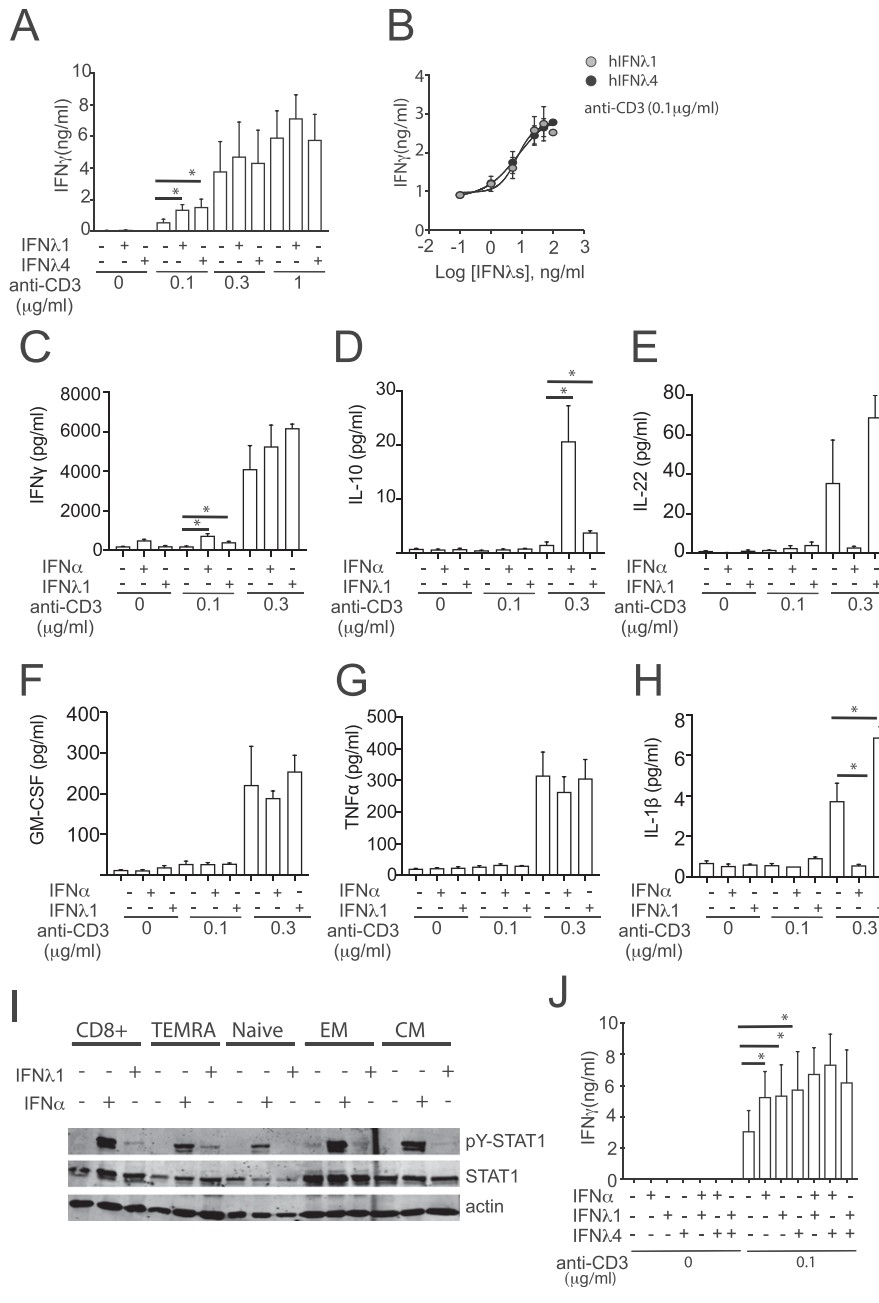

**Figure 4. IFNλ4 provides a co-stimulatory role during CD8+ T-cell activation.**
**(A, B)** CD8+ T cells were plated for 48 h either (A) on a 96-well plate pre-coated with three different concentrations of anti-CD3 (0.1; 0.3 and 1 μg/ml) in the presence of IFNα (1,000 IU/ml), IFNλ1 (100 ng/ml), or IFNλ4 (100 ng/ml) or (B) on a 96-well plate pre-coated with 0.1 μg/ml of anti-CD3 in the presence of increasing doses of IFNλ1 and IFNλ4. Released IFNγ was determined by ELISA. Data are representative of three independent experiments. **(C, D, E, F, G, H)** CD8+ T cells were plated for 48 h either on a 96-well plate pre-coated with three different concentrations of anti-CD3 (0.1 and 0.3 μg/ml) in the presence of IFNα (1,000 IU/ml) or IFNλ1 (100 ng/ml). **(C, D, E, F, G, H)** Released IFNγ (C), IL-10 (D), IL-22 (E), GM-CSF (F), TNFα (G), and IL-1β (H) were determined using Meso Scale Discovery assay. Data are representative of three independent experiments. **(I)** Total CD8+ and purified TEMRA, naïve, EM, and CM CD8+ T-cell populations were stimulated with IFNα (1,000 IU/ml) or IFNλ1 (100 ng/ml) for 15 min. Phosphorylated STAT1 (pY-STAT1), total STAT1, and actin protein were analyzed by Western blotting using total cellular extracts. A representative blot from two independent experiments is shown. **(B, J)** CD8+ T cells were treated with individual or combinations of IFNα (1,000 IU/ml), IFNλ1 (100 ng/ml), and IFNλ4 (100 ng/ml) for 48 h in the presence or absence of anti-CD3 mAbs stimulation and analyzed as described in (B). **(A, D)** Mean ± SEM from three independent experiments (n = 3 for A, D) are shown. *P < 0.05, **P < 0.01, and ***P < 0.001 (paired t test).

detecting 15 markers in T cells isolated from the liver of nine ΔG and four TT CHC patients.

A Wilcoxon test was used to identify differences in the number of cells with a positive phenotype of each surface marker used in the FACS analysis (Fig 6A). These studies showed lack of significant difference in any marker, indicating that there was no enrichment for any one specific population between ΔG and TT patient genotypes when individual markers were considered. As PD1 marker was present in this panel, the flow cytometry data confirmed the histological data for this surface protein.

A further analysis was made to determine the presence of unique T-cell populations defined by co-expression of different markers. A large-data optimized tSNE algorithm, opt-SNE (Belkina et al, 2019 *Preprint*), was used to embed the high-dimensional FACS data into a low dimensional space amenable to density and phenotype clustering using a combination of DBScan and binary clustering methods (Hahsler & Piekenbrock, 2018; Spagnuolo, 2018). This analysis showed the presence of a large number of clusters, some of which were either represented by a sole patient, or were not enriched in cells from either IFNλ4 producers (ΔG) or non-producers (TT) and thus could not reveal any IFNλ4-genotype-dependent phenotypes. To identify phenotypically distinct clusters that were enriched for either the TT or ΔG phenotype and representative of all patients with either IFNλ4-genotype, we used a

A

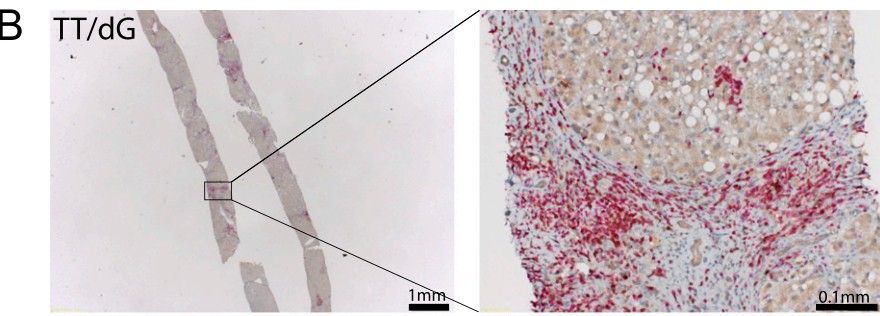

**Figure 5.    Detection of CD3⁺/PD1⁺ cells in the liver of chronic hepatitis C patients.**
**(A)** Paraffin-embedded liver biopsies were stained using mAbs specific for CD3 and PD1 (A) Frequency of CD3⁺/PD1⁺ cells in the liver of chronic hepatitis C patients with ΔG and TT IFNλ4 genotype. **(B, C)** Representative bright-field images of CD3 and PD1 staining of liver sections of a patient of TT/ΔG genotype and (C) TT/TT genotype. Red, CD3 signal; brown, PD1 signal.

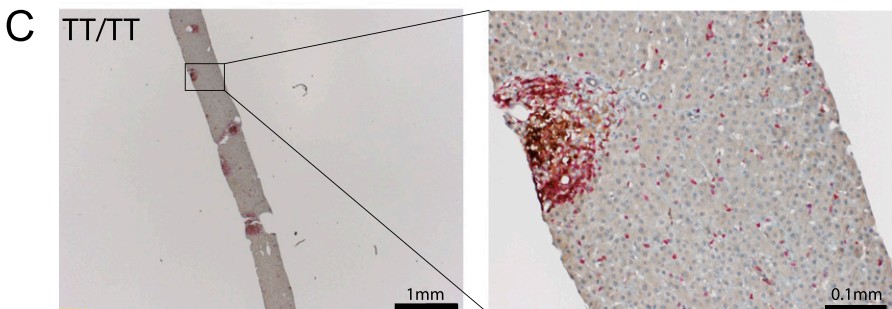

Poisson ratio test which was additionally controlled for patient equality and diversity using the Gini-coefficient and Gini–Simpson index, respectively (Fig 6B and C). Clusters whose IFNλ4 ΔG/TT genotype ratio was in the top and bottom 10th percentile and had a false discovery rate of >0.001 were considered to be significantly enriched. This resulted in 100 phenotypically distinct clusters (Fig 6D) that are broadly clustered by expression of CD8, 2B4, KLRG1, CD127, and CD57 markers.

From these TT and ΔG genotype-enriched clusters, two main groups were immediately apparent with opposing CD57 and CD127 phenotypes. The CD57⁺ CD127⁻ group was constituted mostly by CD8⁺ KLRG1⁺ 2B4⁺ TT-enriched clusters (Fig 6D annotation 1). Both CD57 and 2B4 markers are indicators of senescence and exhaustion in CD8⁺ T cells, respectively (Brenchley et al, 2003; West et al, 2011). Together with the CD45RO⁻ CD127⁻ and KLRG1⁺ phenotypes in this group, these findings indicate that IFNλ4 non-producing patients display larger numbers of T cells with senescent and exhausted phenotypes typical of chronic immune activation in response to viral infection.

In contrast, the CD57⁻ CD127⁺ group contained two CD8⁺ subgroups; CD8⁺ CD45RO⁻ and CD8⁺ CD45RO⁺, both of which had an even split between TT- and ΔG-enriched clusters (Fig 6D annotations 2, 3). The expression of both KLRG1 and 2B4 within these two groups, although not entirely absent, was markedly reduced than

that seen in the CD8⁺ CD57⁺ group. Taken together, this analysis suggests a preponderance for CD8⁺ TT cells to enter a senescent or exhausted phenotype upon repetitive activation.

To further explore the progressive nature of the expression of markers that describe their differentiation states, TT and ΔG cells were ordered according a pseudotime based on the expression of CD57, KLRG1, PD1, and TIM3 markers using a diffusion mapping algorithm (Angerer et al, 2016). Significantly enriched clusters were split into five groups for each IFNλ4-genotype including: total CD8⁺, CD8⁺ CD161⁺, CD8⁺ CD161⁻, CD4⁺, and CD8⁻ CD4⁻ (DN) subpopulations. CD161 was chosen as it marks an important population of T cells in the liver with unique transcriptional and functional phenotype (Northfield et al, 2008; Fergusson et al, 2014).

By examining the distribution of cells along the pseudotemporal axis, it was apparent that TT CD8⁺ cells were predominantly located at the end of the pseudotemporal ordering in a CD57⁺ KLRG1⁺ state independently of their CD161 phenotype (Figs 7 and 8). In contrast, total ΔG CD8⁺ cells were predominantly distributed at the beginning in a KLRG1⁻ CD57⁻ CD127⁺ state (Figs 7 and 8).

Closer analysis of the CD8⁺ population by splitting into CD161⁺ and CD161⁻ populations revealed exceptions to this general observation. The CD8⁺ CD161⁻ TT population showed distribution of cells in both early and late pseudotime, whereas the CD8⁺ CD161⁻ ΔG

**Table 1.  Characteristic of patients included in the study.**

| Nr | Gender | Age at biopsy | rs815 (IFNL4) | HCV genotype | $Log_{10}$ (viral load) | Metavir | FACS | Immunohistochemistry |
|---|---|---|---|---|---|---|---|---|
| 1 | f | 64 | ΔG/ΔG | 1b | 5.44 | A2/F1 | X | X |
| 2 | m | 58 | ΔG/ΔG | 3a | 5.47 | A3/F3 | X | X |
| 3 | m | 24 | ΔG/ΔG | 1 | 4.81 | A1/F0 | X | X |
| 4 | m | 49 | ΔG/ΔG | 4 | 6.50 | A1/F1 | | X |
| 5 | m | 61 | ΔG/ΔG | 1b | n.a | A2/F2 | | X |
| 6 | m | 51 | ΔG/ΔG | 1b | 5.75 | A2/F4 | | X |
| 7 | m | 40 | ΔG/ΔG | 3a | 5.66 | A1/F1 | | X |
| 8 | f | 53 | ΔG/ΔG | 4a/c/d | 5.90 | A3/F3 | | X |
| 9 | m | 48 | ΔG/ΔG | 4a/c/d | 6.03 | A2/F2 | | X |
| 10 | m | 30 | ΔG/ΔG | 1a | 5.03 | A1/F1 | | X |
| 11 | m | 24 | ΔG/ΔG | 1a | 5.26 | A1/F0 | | X |
| 12 | m | 52 | TT/ΔG | 3a | 6.69 | A2/F2 | X | X |
| 13 | m | 56 | TT/ΔG | 3a | 5.75 | A3/F4 | X | X |
| 14 | m | 50 | TT/ΔG | 1a | 6.92 | A3/F2 | X | X |
| 15 | f | 37 | TT/ΔG | 1a | n.a | A1/F2 | X | X |
| 16 | m | 46 | TT/ΔG | 4a/c/d | 5.06 | A3/F4 | X | X |
| 17 | m | 45 | TT/ΔG | 1b | 5.25 | A1/F1 | X | X |
| 18 | M | 15 | TT/ΔG | 1a | 5.44 | A1/F1 | | X |
| 19 | f | 47 | TT/ΔG | 1a | 6.78 | A2/F4 | | X |
| 20 | m | 45 | TT/ΔG | 1a | 6.21 | A2/F2 | | X |
| 21 | m | 42 | TT/TT | 3a | 6.98 | A2/F1 | X | X |
| 22 | f | 52 | TT/TT | 4 | n.a | A2/F2 | X | X |
| 23 | m | 77 | TT/TT | 1b | 5.92 | A2/F3 | X | X |
| 24 | m | 61 | TT/TT | 1a | n.a | A2/F1 | X | X |
| 25 | m | 43 | TT/TT | 1 | 5.97 | A1/F1 | | X |
| 26 | f | 45 | TT/TT | 3a | 5.06 | A1/F1 | | X |
| 27 | f | 45 | TT/TT | 1b | 6.38 | A2/F1 | | X |
| 28 | m | 53 | TT/TT | n.a | 6.47 | A2/F2 | | X |
| 29 | f | 52 | TT/TT | 1a | 6.09 | A2/F1 | | X |
| 30 | m | 36 | TT/TT | 3a | 5.90 | A1/F1 | | X |
| 31 | m | 43 | TT/TT | 1a | 3.60 | A1/F1 | | X |
| 32 | m | 39 | TT/TT | 3a | 6.41 | A1/F1 | | X |
| 33 | f | 41 | TT/TT | 1b | 6.44 | A2/F3 | | X |
| 34 | m | 50 | TT/TT | 3a | 6.52 | A2/F1 | | X |
| 35 | m | 53 | TT/TT | 3a | 6.52 | A3/F4 | | X |
| 36 | m | 56 | TT/TT | 1a | 6.80 | A3/F3 | | X |
| 37 | m | 45 | TT/TT | 3a | 5.87 | A2/F4 | | X |
| 38 | f | 37 | TT/TT | 1a | 6.19 | A2/F3 | | X |
| 39 | m | 53 | TT/TT | 4 | 6.12 | A1/F1 | | X |

population was preferentially distributed in early pseudotime (Fig 8). Furthermore, the CD8$^+$ CD161$^-$ ΔG population showed decrease in CD127 expression correlating with the increase in expression of CD57, KLRG1, 2B4, and PD1. Taken together, these findings indicated a limited accumulation of cells with senescent/exhaustion phenotype in ΔG patients.

Important differences were observed also in the CD8$^+$ CD161$^+$ populations. Whereas CD8$^+$ CD161$^+$ ΔG and TT cells displayed almost

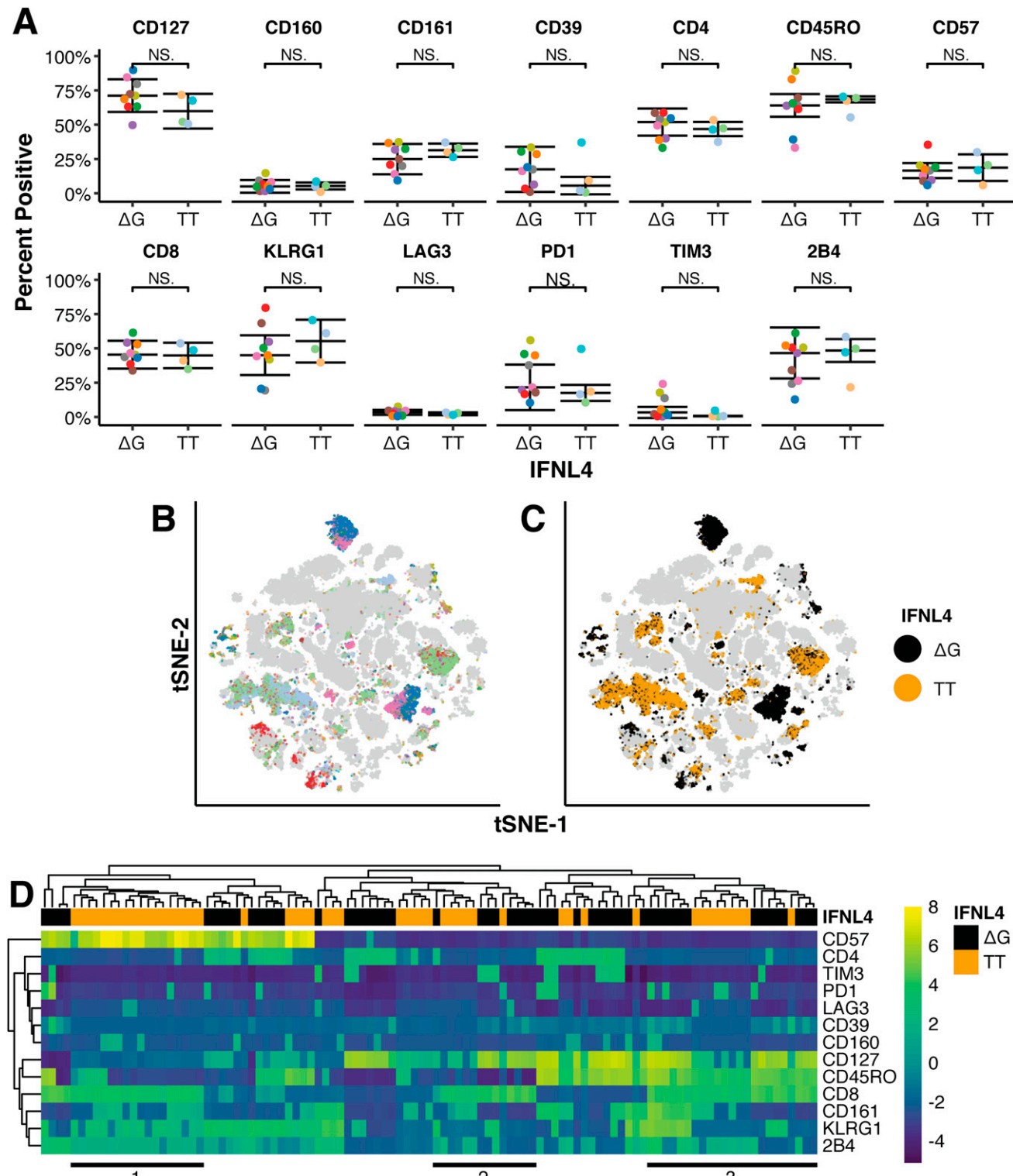

**Figure 6. Phenotypic characterization and multidimensional analysis of patient liver biopsies.**
Intrahepatic lymphocytes (IHLs) were isolated from fresh liver biopsy tissue obtained from chronically hepatitis C virus–infected patients either carrying the IFNλ4 ΔG or TT allele. IHLs were subjected to multicolor FACS analysis. **(A)** *t* tests identified no significant differences (*P* > 0.05) in the percent of cells with a positive phenotype for any cell surface marker between patients with ΔG (n = 9) or TT (n = 4) IFNλ4 alleles. **(B, C)** tSNE dimensional reduction and clustering enabled identification of clusters enriched for either IFNλ4 allele (C) using a Poisson test (false discovery rate < 0.001; colored dots) while maintaining within-cluster representation of patients with each IFNλ4 allele (B). Cells not selected for further analysis are gray. **(B, C, D)** Hierarchical clustering of the median marker expression of selected clusters from (B) and (C), revealing three groups (1, 2, 3) of CD8⁺ clusters with opposing expression of exhaustion and senescence markers CD57, CD127, KLRG1, PD1, and 2B4.

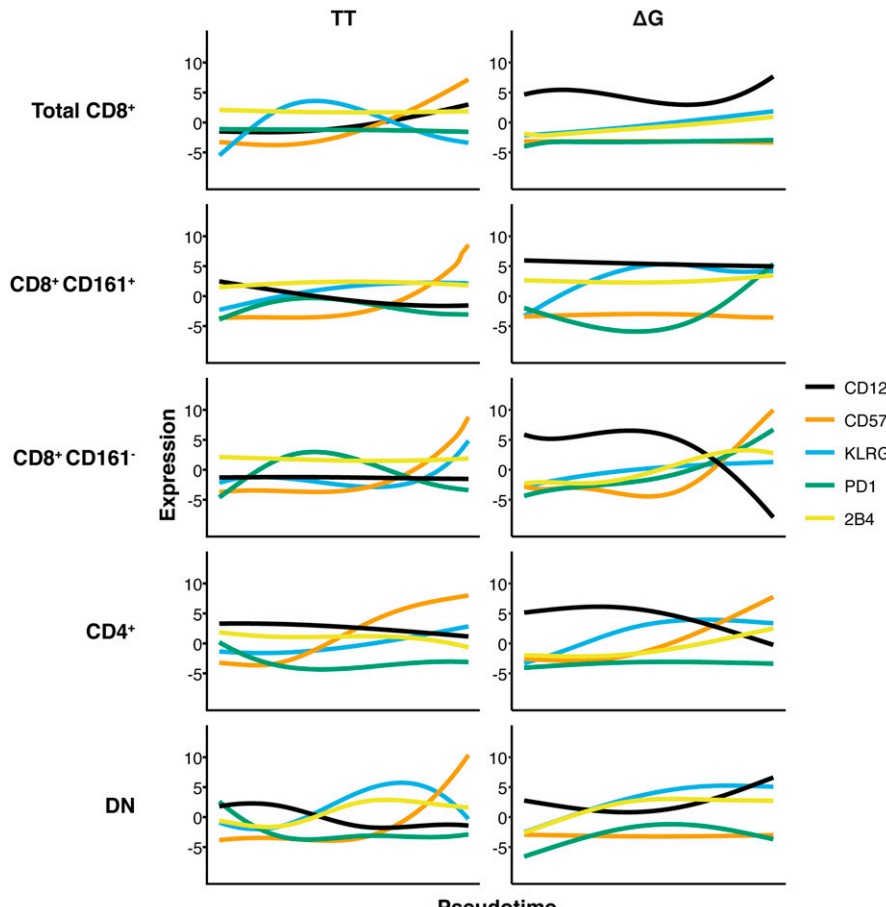

**Figure 7. Pseudotime ordering of CD8⁺, CD4⁺, and double-negative cell populations indicates differentiation state-dependent changes in exhaustion and senescence markers.**
IFNλ4 allele enriched clusters were embedded by diffusion mapping and temporally ordered based on the co-expression of CD57, KLRG1, PD1, and TIM3. Curves fitted to the temporally ordered cells indicate smoothed average of marker expression.

similar temporal distribution (Fig 8), in TT cells, expression of CD127 negatively correlated to the expression of CD57 and KLRG1 (Pearson = −0.66 and −0.50, respectively). That is, these cells are CD127-highly positive at the beginning of the temporal ordering and become CD127-negative as expression of KLRG1 and CD57 increases. In contrast, the CD8⁺ CD161⁺ ΔG cells did not show a pseudotime-dependent relationship between the expression of CD127, CD57, and KLRG1; showing no pseudotime-dependent change in CD127 or CD57 phenotypes across the temporal axis, which remained high and low, respectively. Nevertheless, this population did show a pseudotime-dependent increase in the expression of KLRG1 and PD1. These findings indicated that in CD8⁺ CD161⁺ ΔG cells, there were fewer cells expressing the exhaustion-associated marker CD57 and more cells expressing the PD1 and CD127 markers, indicating previous cell activation (Larbi & Fulop, 2014).

The analysis also showed that there were no major differences between CD4⁺ TT and CD4⁺ ΔG populations, which mostly distributed early in pseudotime and displayed expression patterns similar to that of the CD8⁺ CD161⁻ ΔG cells.

CD4⁻ CD8⁻ DN TT and ΔG cells were temporally distributed towards the end of the continuum, similarly to the distribution of CD8⁺ CD161⁺ TT and ΔG cells. However, only the double-negative TT cells displayed a pseudotime-dependent increase in CD57 and KLRG1 expression correlating with a decrease in CD127 expression. The DN ΔG cells remained CD127⁺ without any change in CD57, despite an increase in KLRG1 over time.

In conclusion, such analysis showed that only CD8 and DN cells differed between IFNλ4-producer (ΔG) and non-producer (TT) patients. In particular, in TT patients the CD8⁺ 161⁻ population showed a phenotype of cells with senescent phenotype, which might reflect repetitive activation during chronic infection. On the contrary, in ΔG patients the same population showed marked tendency to remain partially activated and without evidence of senescence/exhaustion.

## Discussion

HCV infection triggers a rapid and strong innate immune response characterized by induction of a large variety of ISGs (Heim & Thimme, 2014). Despite activation of this important arm of innate immunity, clearance of acute HCV infection depends on activation of virus-specific CD4 and CD8 T cells (Grakoui et al, 2003; Shoukry et al, 2003). HCV clearance is also facilitated by a genetic polymorphism that prevents production of active IFNλ4. Indeed, the wild-type allele of *IFNL4* gene that encodes a functional IFNλ4 protein is associated with inefficient adaptive immune responses and development of chronic HCV infection (Terczynska-Dyla et al, 2014). The link between *IFNL4* genotypes and the cellular immune response against HCV remains to be determined. As the IFNLR was found mostly expressed by epithelial cells (Sommereyns et al,

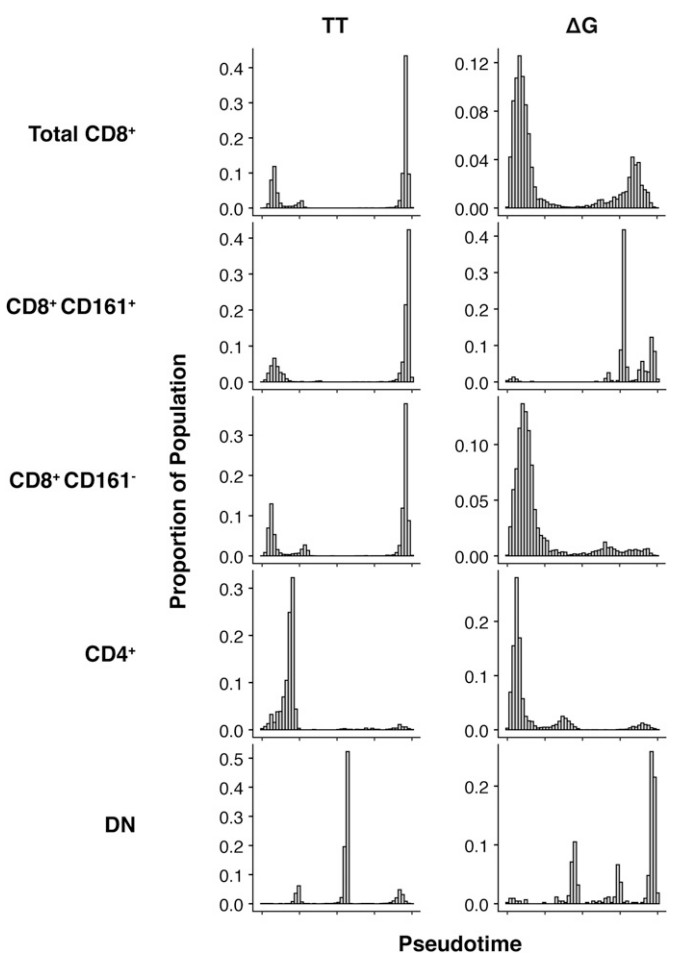

**Figure 8. Distribution of cells across pseudotime.**
The proportion of cells is differentially distributed across pseudotime in a lineage-dependent manner. Total CD8⁺ TT cells (n = 14,081) are distributed late in the temporal ordering, whereas total CD8⁺ ΔG cells (n = 6,230) are distributed earlier. No difference was observed in the temporal distribution of CD8⁺ CD161⁺ TT or ΔG cells (n = 4,226 and 513, respectively). In contrast, an IFNλ4-dependent differential distribution of cells was observed in CD8⁺ CD161⁻ TT and ΔG cells (n = 9,855 and 5,717, respectively). CD4⁺ TT and ΔG cells (n = 1,591 and 6,959, respectively) were temporally distributed similarly to CD8⁺ TT cells. Double-negative (DN) TT and ΔG cells (n = 6,722 and 437, respectively) were distributed similarly to CD8⁺ CD161⁺ cells.

2008), it remains also unclear whether IFNλ regulate immune response in a direct or indirect manner. Several immunomodulatory activities of IFNλ1-3 have been described, including the up-regulation of IL-6, IL-8, and IL-10 production in PBMCs (Jordan et al, 2007a), alteration of the Th1/Th2 T-cell balance (Jordan et al, 2007b), reduction of IL-13 production by T cells (Srinivas et al, 2008), or induction of ISGs in B cells (de Groen et al, 2015) and in plasmacytoid dendritic cells (Kelly et al, 2016). It remains unclear whether also IFNλ4 has immunomodulatory capacities (Heim et al, 2016).

In the present study, we investigated whether different IFNλs affect B- and T-cell functions, and in particular whether IFNλ4 has immunosuppressive functions, which could support its inhibitory role in the adaptive host response to HCV infection. Using STAT1 activation as readout for IFNλ stimulation, we directly assessed cell activation through IFNLR triggering. We found that only B and CD8

T cells, isolated from circulating blood of healthy donors, respond to IFNλ. The strongest STAT1 phosphorylation was detected in B cells and the strength of STAT1 activation in different lymphocytic populations correlated with the *IFNLR1* transcripts. These findings are consistent with other studies reporting that B cells express the highest levels of *IFNLR1* mRNA among PBMCs (Witte et al, 2009). In addition, and consistent with a previous report (Rivera, 2019), we found neutrophils to be responsive to IFNλ.

Further characterization of the IFNλ-responsive B-cell populations revealed that both CD27-negative cells (enriched in naïve populations) and CD27-positive cells (enriched in memory B cells) responded to IFNλ. In contrast, among CD8⁺ T cells, effector memory cells (TEMRA) were the most responsive followed by central memory cells, whereas naïve cells were nonresponsive. CD4 T-cell populations instead did not show STAT1 phosphorylation upon incubation with IFNλ. Monocytes did not respond to IFNλ probably caused by a very low expression of the IFNLR1, as previously reported (Mennechet & Uze, 2006; Liu et al, 2011). When CD14 monocytes were differentiated into dendritic cells, they acquired IFNλ responsiveness, as previously reported (Mennechet & Uze, 2006).

Having identified the lymphocyte populations reacting to IFNλ, we studied several possible immunosuppressive effects. We detected no differences of JAK-STAT signaling in B cells and CD8 T cells after stimulation by IFNλ1 and IFNλ4, thus suggesting that IFNλ4 acts as a *bonafide* IFN in this regard, similarly to other type I IFNs. IFNλ4 also did not interfere with the other IFNs using different activation assays. IFNλ exerted minimal direct effects on B-cell activation as measured by changes in CD69 expression levels, despite expression of IFNRL. It also showed synergistic effects with TLR9 stimulation of B cells, although with lower efficacy than other type I IFNs.

In another series of experiments, we studied whether IFNλ4 interferes with antibody production by B cells. Most of HCV-infected patients produce antibodies against structural and non-structural HCV proteins during the acute phase of infection (Logvinoff et al, 2004). Whether these antibodies participate in HCV clearance remains unclear. We did not find inhibitory activity of IFNλ4 on TLR9 induced IgG production, confirming that IFNλ4 is similar to other tested IFNs also in this regard.

Like other IFNs, IFNλ4 increased the release of the immunosuppressive IL-10 cytokine, when B cells were co-stimulated with TLR9 ligand. However, even larger increased IL-10 production was induced by IFNα and IFNλ1, thus excluding that this mechanism is unique to IFNλ4.

Both IFNλ1 and IFNλ4 promoted the release of IFNγ by T cells upon suboptimal stimulation with anti-CD3 monoclonal antibodies. Similarly, IFNλ stimulation promoted the secretion of anti-inflammatory cytokines IL-10 and IL-22. Surprisingly, stimulation in the presence of IFNλ4 and IFNα exerted different effects on secretion of IL-1β, with a significant increase of this pro-inflammatory cytokine observed only with IFNλ. Although it is known that type I IFNs have anti-inflammatory properties because they inhibit IL-1β secretion (Huang et al, 1995; Billiau, 2006), little is known on the effects of type III IFN on IL-1β secretion. One study reported that IFNλ drives a pro-inflammatory phenotype during monocytes differentiation, increasing the secretion of inflammatory mediators including CCL2, IL-1β, and TNF (Read et al, 2019). IL-1β promotes inflammation by inducing the expression of proinflammatory genes, by recruiting immune cells to the site of

infection, and by modulating infiltrating cellular immune-effector action. Interestingly, CHC patients exhibited elevated levels of serum IL-1β compared with healthy controls (Negash et al, 2013). Further studies are required to understand the mechanism of IFNλ regulation of IL-1β secretion.

The most responsive populations were TEMRA and CM CD8 T cells, whereas naïve CD8 T cells were non-responsive. These findings were in accordance with the STAT1 phosphorylation studies and with detected up-regulation of IFNR genes, but do not explain the immunosuppressive effects of IFNλ4. They indicate that type I IFNs sustain the effector functions of primed and antigen-responding CD8 T cells, whereas they probably have minor effects during T-cell priming with antigen.

Another reported effect of stimulation with IFNα is induction of inhibitory receptors in T cells (Francisco et al, 2010; Terawaki et al, 2011). Therefore, we attempted to correlate the presence of CD3⁺PD1⁺ cells in the liver of HCV patients with IFNλ4-producer and non-producer genotype. These immunohistochemistry studies revealed the presence of PD1-expressing T cells, whose numbers, however, were not significantly different in the two groups of patients.

Significant differences were instead observed using multicolor flow cytometry. Using tSNE analysis we identified several clusters of T-cell populations. These clusters were further selected for their representation in all patients and classified for enrichment in either TT or ΔG donors. Major differences were detected in the CD8 and DN populations, but not in CD4 cells, indicating alterations only in cell populations which reacted to IFNλ4 in vitro. When cells were distributed using a pseudotime order, together with expression of the CD161 marker, cells from TT or ΔG donors differed in multiple ways. Indeed, CD8⁺ CD161⁻ ΔG cells showed preferential distribution in the early phase of pseudotime and did not express the PD1, KLRG1, 2B4, and CD57 markers. Instead, they showed high expression of CD127, a marker mostly expressed by central memory T cells and positively correlating with expression of CD28 (Larbi & Fulop, 2014). As CD127 participates in the regulation of host cell homeostasis, proliferation of differentiated T cells, and also in cell survival, these cells might represent a potentially still active population without evidence of senescence or exhaustion (Larbi & Fulop, 2014). On the contrary, the minor population of CD8⁺ CD161⁻ ΔG cells distributing in the late pseudotime dimension resembled the corresponding population from TT donors. These T cells expressed high levels of PD1, KLRG1, 2B4, and CD57, thus resembling senescent/exhausted cells. The reason of the accumulation in IFNλ4-producers of CD8 T cells with central memory-like and not senescent phenotype is intriguing. It is tempting to speculate that as the affected T-cell populations are the same showing direct response to IFNλ4 in vitro, the effect is T-cell intrinsic. As CD8⁺ CD161⁻ cells belong to the classical population of adaptive T cells, these cells might become low responders to the antigen, thus proliferating slowly in the presence of IFNλ4. This effect could be obtained by direct interference of IFNLR signaling with antigen responsiveness. Indeed, during LCMV infection in mice, STAT1 activation in T cells induced inhibition of CD8 cell proliferation (Gil et al, 2006). Future studies will address the molecular mechanisms of IFNλ4 on CD8⁺ CD161⁻ cells.

The second affected population was represented by CD8 cells expressing CD161. These cells are abundant in the liver and in other mucosal tissues. They frequently recognize non–MHC-restricted antigens, express chemokine receptors associated with tissue homing in resting and inflammatory conditions (Billerbeck et al, 2010), and proliferate in the presence of IL-12 and IL-18 independently of TCR stimulation (Ussher et al, 2014). These cells were found mostly in the late pseudotime and only in ΔG donors. They showed high expression of PD1, KLRG1, CD127, and lack of CD57. These cells therefore represent a population which is not exhausted, although it expresses typical markers of TEMRA cells (Larbi & Fulop, 2014). As for the CD8⁺ CD161⁻ cells, also in this case, it remains unknown the molecular mechanism leading to the accumulation in ΔG donors of cells with these unique phenotypes. As speculated with CD8⁺ CD161⁻ cells, it could be that IFNλ4 reduces antigen responsiveness, thus limiting CD57 expression, and without limiting antigen-independent proliferation after stimulation with IL-12 and IL-18. Indeed, expression of CD57 has been associated with persistent antigenic stimulation and critically shortened telomeres (Brenchley et al, 2003) and does not occur after proliferation induced by IL-12 or IL-18 (Kurioka et al, 2018).

In contrast with other reports (Srinivas et al, 2008), our data demonstrate that CD4 cells do not respond to IFNλ probably because of low IFNLR expression according to our PCR data. Although CD4 cells are not responsive to IFNλ stimulation, they could be indirectly modulated by IFNλ4 through its effects on B and CD8 T cells, or dendritic cells. Clarification of these possibilities would require detailed analysis of intrahepatic lymphocytes isolated from a large cohort of CHC patients because IFNλ4 gene is not expressed in mice (Wack et al, 2015).

In conclusion, our results show that IFNλ4 displays typical IFN activities and acts directly on unique populations of immune cells. The same populations of T cells are affected in the liver of IFNλ4-producer donors indicating T-cell–intrinsic effects, which lead to limited senescence after antigen stimulation.

# Materials and Methods

### Healthy blood donors

Blood samples were obtained from Regional Blood Transfusion Service, Swiss Red Cross, Basel, from healthy male and female donors (18–65 yr old) as buffy coats. All donors gave written informed consent.

### Human liver biopsies and DNA isolation

Liver biopsies and blood samples (EDTA-anticoagulated blood) were obtained from HCV-infected patients (n = 13) (patients characteristics are shown in Table 1) in the outpatient clinic of the Division of Gastroenterology and Hepatology, University Hospital Basel, Switzerland. Biopsy material that was not needed for routine histopathology was used for research purposes after obtaining written informed consent. The use of biopsy material for this project was approved by the Ethikkommission Nordwest- und Zentralschweiz, Basel, Switzerland, protocol number M989/99. Total DNA was isolated from blood using DNeasy Blood & Tissue Kit (QIAGEN) according to the manufacturer's instructions. IFN-λ4 genotype was determined as described previously (Terczynska-Dyla et al, 2014).

## Isolation and stimulation of PBMCs and liver infiltrating lymphocytes

PBMCs were isolated from blood samples by standard density-gradient centrifugation protocols (SEcoll Human; Seraglob), washed twice in PBS, and analyzed immediately or cryopreserved in freezing medium (90% fetal bovine serum and 10% dimethyl sulfoxide). PBMCs were stimulated with 100 ng/ml IFNλ1 and as control with 1,000 IU/ml IFNα. Total cellular proteins were isolated after 15 min of IFN stimulation and the degree of STAT1 phosphorylation (pY-STAT1) was determined by Western blotting. Liver biopsy samples were collected in PBS and then washed twice again with PBS to remove cell debris and red blood cells. Isolation of liver-infiltrating lymphocytes was performed by mechanical disruption of the tissue through a 40-$\mu$m cell strainer (Falcon) using a 5-ml syringe plunger (CODAN Medical ApS). The mesh was rinsed twice with PBS to ensure maximal recovery. Infiltrated liver immune cells were cryopreserved in freezing medium.

## Antibodies and reagents

Antibodies, chemicals, and kits are described in Table S1.

## Isolation of cell subpopulations from PBMCs

PBMCs were isolated as described in the previous section. Monocytes, CD3$^+$, CD8$^+$ T, CD4$^+$ T, and B cells were enriched by positive selection (>90% purity) using magnetic beads, according to the manufacturer's instructions (Miltenyi Biotec). Cells were rested for 2 h in complete medium (RPMI-1640, 10% fetal bovine serum, 50 U/ml penicillin and 50 $\mu$g/ml streptomycin). Mo-DCs were differentiated in vitro from purified CD14$^+$ monocytes in the presence of 800 U/ml GM-CSF and 500 U/ml IL-4 as described previously (Nair et al, 2012). CD16$^+$/CD3$^-$ NK cells were isolated from PBMCs stained with anti-CD3 and anti-CD16 antibodies by FACSs. Cell sorting was performed with a BD influx cell sorter (BD Bioscience). For sorting of naïve, TEMRA, EM, and CM CD8$^+$ T cells, CD8$^+$ T cells were stained with anti-CCR7 and anti-CD45RA (BD Bioscience) monoclonal antibodies (mAbs). Naïve, EMRA, EM, and CM CD8$^+$ T cells were identified as CD45$^+$/CCR7$^+$, CD45$^+$/CCR7$^-$, CD45$^-$/CCR7$^-$, and CD45$^-$/CCR7$^+$, respectively. Cells were rested in complete medium for 2–4 h at 37°C before experimentation. Naïve (CD19$^+$/CD27$^-$) and memory (CD19$^+$/CD27$^+$) B cells were isolated from CD19$^+$ B cells using CD27-capturing beads according to the manufacturer's instructions (Miltenyi Biotec).

## Isolation of neutrophils and migration assay

Neutrophils were isolated from blood samples over a 62.5% Percoll gradient (GE Healthcare) in Ca$^{2+}$- and Mg$^{2+}$-free HBSS as previously described (Mocsai et al, 2000) with a purity of more than 90%. $5 \times 10^5$ neutrophils were placed into a 24-well PET Transwell (8 $\mu$m pore membrane). As positive control of 100 nM N-formylmethionine-leucyl-phenylalanine (fMLP; Sigma-Aldrich) was added to the medium placed on the bottom of the wells. IFNλ1 or IFNλ4 were used as stimulus alone or in combination with fMLP. Neutrophils were allowed to migrate for 4 h. After 4 h, the medium was collected, and migrated neutrophils were counted using FACS. Data are reported of percentage of migrated neutrophils over the total number of neutrophils.

## Total RNA extraction and quantitative PCR

Total RNA from PBMCs was isolated using NucleoSpin RNA (Macherey-Nagel AG) according to the manufacturer's instructions. cDNA was synthesized from 400 ng of total RNA using MultiScribeTM Reverse Transcriptase (Applied BiosystemsTM) and random hexamer primers in a 25-$\mu$l reaction. For all samples, "-RT" controls (reactions omitting the reverse transcriptase) were performed. Real-time qRT-PCR was performed using FastStart Universal SYBR Green Master (Roche Diagnostics AG) or TaqMan Universal PCR Master Mix No AmpErase UNG (Thermo Fischer Scientific) using an ABI 7500 detection system (Applied Biosystems, Thermo Fisher Scientific). Primers and probes are listed in Table S2. The specificity of the PCR products was assessed on a 3% agarose gel and sequenced. Gene transcript expression levels were calculated using the ΔΔCT method relative to *GAPDH*.

## Whole-cell lysates and immunoblots

Whole-cell lysates and immunoblots were prepared and performed as described previously (Duong et al, 2004) using the following antibodies: Phospho-STAT1 (Tyr701) and $\beta$-actin and STAT1 (N-term).

## T-cell activation and cytokines measurement

Human CD8$^+$ T cells were activated with plate bound anti-CD3 (OKT3; BioLegend) mAbs. 96-well flat bottom plates were coated with anti-CD3 mAbs at different concentrations (1 $\mu$g/ml; 0.5 and 0.25 $\mu$g/ml) and incubated overnight at 4°C. Then the coating solution was removed and CD8$^+$ T cells were added at $3 \times 10^5$ cells/well. 48 h after activation, cell supernatants were harvested and IFN-γ was measured using an IFN-γ ELISA. For IFN-γ measurement, 96-well plates were coated with 2.5 $\mu$g/ml of anti-IFNγ (BioLegend) antibody and incubated overnight at 4°C. The coating solution was removed and supernatant from activated B cells was added and incubated 2 h at room temperature. Biotin anti-IFNγ antibody (1 $\mu$g/ml) together with HRP Streptavidin (dilution 1/2,000) was used for IFNγ detection.

In addition, supernatant was used to measure the secretion of IFN-γ, IL-22, IL-10, IL-1B, TNFα, and GM-CSF using U-PLEX Human 4 and 7 plex Biomarker Group 1 (Meso Scale Discovery) following the manufacturer's instructions.

## B-cell activation

CD19$^+$ cells were stimulated with 1,000 IU/ml IFNα or 100 ng/ml IFN-λ1 or IFNλ4 (kindly provide by Prof R Hartmann) alone or combined with CpG ODN2006 at 0.8 or 2.5 $\mu$g/ml for 48 h. Cellular activation and surface marker expression was measured by FACS. IgG and IL-10 protein production in the culture supernatants was assessed by ELISA. IgG was measured in the supernatants using sandwich ELISA kits specific for total IgG (detection limit of 1.6 ng/ml) following the manufacturer's (Thermo Fisher Scientific) instructions. For IL-10 measurement, 96-well plates were coated with 2.5 $\mu$g/ml of anti-

IL-10 (BioLegend) antibody and incubated overnight at 4°C. The coating solution was removed and supernatants from activated B cells were added and incubated for 2 h at room temperature. Biotinylated anti-IL-10 antibodies (1 μg/ml) and HRP-Streptavidin (dilution 1/2,000) were used for IL-10 detection.

### Flow cytometry

PBMCs were subjected to surface staining to identify lymphocyte populations using the mAbs listed in Table S1. Cells were resuspended in staining buffer (PBS containing 0.02% NaN$_3$ and 0.5% human albumin) and stained for 20 min at 4°C in the dark. Data were acquired using a BD Accuri C6 (BD Bioscience) or CytoFLEX (Beckman) flow cytometer and analyzed with Flowjo 10.1.r1 (Tree Star). For intracellular staining, cells were fixed and permeabilized with Perm Buffer (BD Bioscience) for 20 min on ice. After washing, cells were incubated with blocking solution (5% BSA in PBS) for 1 h at room temperature. After removal of the blocking solution, cells were incubated with pY-STAT1 antibody or rabbit IgG as control at 4°C overnight. Next day cells were washed and incubated with AlexaFluor 647–labeled goat antirabbit Ig for 1 h at room temperature. After washing, cells were resuspended in PBS and analyzed using BD Accuri C6 (BD Bioscience). Data were analyzed with Flowjo 10.1.r1 (Tree Star).

### Lymphocyte phenotype analysis

Paired intrahepatic lymphocytes and PBMCs were thawed in PBS with 1% DNase to prevent cell clumping. After centrifugation, the cells were resuspended in complete medium and rested for 1 h at 37°C before staining. Cells were washed with PBS/1% DNase and resuspended in blocking buffer (PBS containing 50% human serum) for 30 min at room temperature. For staining, the cells were resuspended in staining buffer (PBS containing 0.02% NaN$_3$ and 0.5% human albumin) and antibodies were added for 20 min at 4°C in the dark. After washing, the cells were resuspended on PBS and analyzed using an LSR Fortessa flow cytometer (BD Bioscience). Data were analyzed using FlowJo v10.r1 (Tree Star). CD3$^+$ living cells were gated and FACS data were exported from FlowJo v10.r1 into the R statistical computing environment and transformed with an inverse hyperbolic sine function (asinh). For further analysis, the transformed data were separated into IHL and PBMC T-cell sets. Each group was subjected to two-dimensional tSNE reduction using the R package fftRtsne version 1.0.1 based on the tSNE implementation described by Linderman et al (2019). IFNλ4 genotype specific T-cell clusters were identified by density based spatial clustering using the DBScan (Hahsler & Piekenbrock, 2018) package version 1.1-3 and the results were plotted using the ggplot2 package. Clusters containing significantly higher ratios of either IFNλ4 allele were identified by performing Poisson tests using generalized linear mixed effect models (lme4 [Bates et al, 2015] package version 1.1-21) with patient IDs as the random variable to account for variability in the total number of cells acquired in each patient. CD4$^+$, CD8$^+$, and double-negative (CD4$^-$CD8$^-$) IFNλ4 TT/ΔG–enriched clusters with false discovery rate < 0.05 were split into groups by their CD161 phenotype before pseudotime analysis using destiny (Angerer et al, 2016) version 2.12.0, and expression of CD57, PD1,

KLRG1, and TIM3 markers were used for pseudotemporal ordering. Finally, curves for pseudotemporally ordered marker expression were fitted using mgcv (Wood, 2011) version 1.8-28.

### CD3 and PD1 immunohistochemistry

Immunohistochemical (IHC) staining for CD3 and PD1 was performed on 4-μm sections of formalin-fixed paraffin-embedded liver biopsy tissue obtained from CHC patients. Immunohistochemical staining was performed on a Benchmark immunohistochemistry staining system (Ventana Medical Systems) using primary anti-CD3 and PD1 antibodies (Roche Diagnostic, Table S1) and the iVIEW-DAB chromogenic system for signal visualization as previously described (Piscuoglio et al, 2012). Specifically, sections were pretreated with CC1 (Ventana Medical Systems) before incubation with the primary anti-CD3 and PD1 antibodies. The staining signal was then revealed using the iVIEW-DAB system. Negative controls omitting the primary antibody were included in each run. Blind scoring of the immunoreactivity was performed independently by an experienced pathologist (LM Terracciano) and two observers (D Calabrese and M Coto-Llerena).

### Statistical analysis

Data are presented as mean value ± SEM. The data were analyzed with Prism4 (GraphPad Software Inc) using a ratio $t$ test. In all analyses, a two-tailed $P < 0.05$ (95% confidence interval) was considered statistically significant. All authors had access to the study data and reviewed and approved the final manuscript.

# Supplementary Information

# Acknowledgements

We thank Hans Henrik Gad and Rune Hartmann for providing recombinant human IFNλ4. We thank Stefan Wieland for scientific assistance and critical revision of the manuscript. Grant support: The work was supported by grants SNF 310030_166202 to MH Heim, and SNF 310030-173240, KFS-3730-08-2015, and PMB-02-17 to G De Libero.

### Author Contributions

M Coto-Llerena: conceptualization, data curation, formal analysis, validation, investigation, methodology, and writing—original draft.
M Lepore: conceptualization, data curation, formal analysis, validation, investigation, visualization, and methodology.
J Spagnuolo: data curation and formal analysis.
D Di Blasi: investigation and methodology.
D Calabrese: investigation and methodology.
A Suslov: investigation and methodology.
G Bantug: conceptualization.
FHT Duong: investigation and methodology.

LM Terracciano: investigation and methodology.

G De Libero: conceptualization, formal analysis, supervision, funding acquisition, validation, and writing—review and editing.

MH Heim: conceptualization, supervision, funding acquisition, validation, project administration, and writing—review and editing.

## Conflict of Interest Statement

The authors declare that they have no conflict of interest.

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
