## [Reviewer comments · Life Science Alliance]

Life Science Alliance

Interferon lambda 4 can directly activate human CD19+ B cells and CD8+ T cells

Mairene Coto-Llerena, Marco Lepore, Julian Spagnuolo, Daniela Di Blasi, Diego Calabrese, Aleksei Suslov, Glenn Bantug, François Duong, Luigi Terracciano, Gennaro De Libero, and Markus Heim
DOI: <https://doi.org/10.26508/lsa.201900612>

Corresponding author(s): Markus Heim and Gennaro De Libero, Basel University

Review Timeline:	Submission Date:	2019-11-25
	Editorial Decision:	2019-12-22
	Revision Received:	2020-09-20
	Editorial Decision:	2020-10-12
	Revision Received:	2020-10-26
	Accepted:	2020-10-26

Scientific Editor: Shachi Bhatt

Transaction Report:

December 22, 2019

Re: Life Science Alliance manuscript #LSA-2019-00612-T

Prof. Markus H Heim
Basel, University
Department of Research
Universitaetsspital Basel
Hebelstrasse 20
Basel, BS 4031
Switzerland

Dear Dr. Heim,

Thank you for submitting your manuscript entitled "Interferon lambda 4 directly activates human CD8+ T cells" to Life Science Alliance. The manuscript was assessed by expert reviewers, whose comments are appended to this letter.

As you will see, the reviewers appreciate your analyses but would expect more support for the biological significance of your findings. Based on the input received, we would like to invite you to submit a revised version of your manuscript to us, addressing the concerns raised. This is in part straightforward, however, please note that we would need strong support from reviewer #2 on such a revised version. Importantly, some conclusions would need backing up via a different approach (rev#2, point 5) and more evidence for the biological significance of your findings would be important to add as well.

Thank you for this interesting contribution to Life Science Alliance. We are looking forward to receiving your revised manuscript.

Sincerely,

B. MANUSCRIPT ORGANIZATION AND FORMATTING:

Reviewer #1 (Comments to the Authors (Required)):

The paper "Interferon lambda 4 directly activates human CD8+ T cells" by Coto-Llerena et al performs a systematic analysis of different immune cells response to IFNL and continues with investigating the CD8+ T cell response in IFNL4 positive and negative patients. First they do provide an highly interesting piece of data, showing no difference in the total amount of T cells between HCV infected patients which are either IFNL4 non producers or IFNL4 producers. Even that this is negative data I think it is an important piece of data. Subsequently, they find an interesting difference in the CD8+, CD161- population, suggesting that HCV infected patients which encode a nonfunctional IFNL4 gene do have a poorer quality of their CD8 T cell response and that these patients are more prone to T cell exhaustion. Even that the molecular mechanism is unclear, this suggests that differences in the CD8+ T cell response can be the reason for the difference in clearance rates between IFNL4 producers and non producers.

Major points

The paper describes CD4+ T cells as non responsive and CD8+ as responsive, based upon figure 1. I do not think the difference between the CD4 and CD8 subset is absolute, either a more careful wording of the conclusions should be used or more definitive data provided.

In figure 2C, IFNL4 leads to substantially less IFI27 induction, suggesting that the activity of this protein prep is less active than the preparation of IFN α and IFNL1, this could be caused by technical difficulties in preparing the IFNL4 protein. Nevertheless, the effect of this on the conclusion drawn should be discussed.

Minor point.

Please check references carefully, some references seem misplaced below are a few examples p. 15 first paragraph "The link between IFNL4 genotypes and the cellular immune response against HCV remains to be determined (Sommerreyns et al., 2008). This reference seems to be misplaced, it is used again in the next sentences where it actually appears to belong.

P18 end of page "Clarification of these possibilities would require detailed analysis of intrahepatic lymphocytes isolated from a large cohort of CHC patients because IFNL4 gene is not expressed in mice (Key et al., 2014)." The reference to Key et al seems out of context.

Reviewer #2 (Comments to the Authors (Required)):

These authors report that CD19+ B cells and, to a lesser degree, CD8+ T cells express IFN-lambda receptors and respond to treatment with recombinant human IFN-lambda 1 and -lambda 4.

1. The authors should consider examining the effects of lambda 1 and lambda 4 on neutrophils because there are several reports showing that neutrophils are IFN-lambda-responsive.
2. The title of this paper ("Interferon lambda 4 directly activates human CD8+ T cells") seems odd in view of the fact that the data provided in Figs. 2 and 3 concern CD19+ B cells instead of CD8+ T cells. I recommend that the authors revise the title as follows: "Interferon lambda 4 can directly activate human CD19+ B cells and CD8+ T cells." Alternatively, if the authors remove Figs. 2 and 3, the current title would be appropriate.

3. Although B cells appear to express IFN-lambda receptors, the results in Fig. 2 show that IFN-lambda exerts minimal direct effects on B cell activation as measured by changes in CD69 expression levels.

4. What is the rationale for choosing IFI27 as the only ISG to confirm IFN responsiveness of B cells in Fig. 2? The authors should measure expression of several, additional, more common ISGs such as MX1, OAS and IFIT1.

5. Based on their results in Fig. 3, the authors conclude that IFN-alpha and -lambda increase production of IL-10 by CpG-stimulated B cells. In Fig. 4, the authors evaluated the effects of IFN-lambda 1 and 4 on IFN-gamma production by CD8+ T cells. IFN-lambda 1 and 4 weakly enhanced IFN-gamma production, but this effect was only observed when the cells were stimulated with a very low concentration of anti-CD3 mAb. The significance of this observation is unclear. T cell receptor activation by anti-CD3 induces expression of multiple cytokines, including IL-10 and IL-22. Therefore, it would be informative to measure the levels of IL-10 and IL-22 produced by T cells after activation with anti-CD3 +/- IFN-alpha and/or IFN-lambda. This would help to broaden the scope of this report.

General comment: It is difficult to gauge the biological significance of the findings described in this paper because the magnitude of responses induced by IFN-lambda 1 and 4 (e.g., phospho-STAT1 activation, changes in CD69 expression, or changes in IL-10 expression) are very weak/modest.

Several previous related studies by others have also examined the ability of various types of blood leukocytes to respond to IFN-lambda. For example:

Freeman J, et al. (2014) Pegylated interferons Lambda-1a and alfa-2a display different gene induction and cytokine and chemokine release profiles in whole blood, human hepatocytes and peripheral blood mononuclear cells. *J Viral Hepat.* 21(6):e1-9.

Yin Z, et al. (2012) Type III IFNs are produced by and stimulate human plasmacytoid dendritic cells. *J Immunol.* 189(6):2735-45.

Kelly A, et al. (2016) Immune cell profiling of IFN- λ response shows pDCs express highest level of IFN- λ R1 and are directly responsive via the JAK-STAT Pathway. *J Interferon Cytokine Res.* 36(12):671-680.

Point by point responses

Reviewer #1 (Comments to the Authors (Required)):

The paper "Interferon lambda 4 directly activates human CD8+ T cells" by Coto-Llerena et al performs a systematic analysis of different immune cells response to IFNL and continues with investigating the CD8+Tcell response in IFNL4 positive and negative patients. First they do provide an highly interesting piece of data, showing no difference in the total amount of Tcells between HCV infected patient which are either IFNL4 non producers or IFNL4 producers. Even that this is negative data I think it is an important piece of data. Subsequently, they find an interesting difference in the CD8+, CD161- population, suggesting that HCV infected patients which encodes a nonfunctional IFNL4 gene do have a poorer quality of their CD8 T cell response and that these patients are more prone to Tcell exhaustion. Even that the molecular mechanism is unclear, this suggests that differences in the CD8+ Tcell response can be the reason for the difference in clearance rates between IFNL4 producers and non producers.

Major points

The paper describes CD4+ Tcells as non responsive and CD8+ as responsive, based upon figure 1 I do not think the difference between the CD4 and CD8 subset are absolute, either a more careful wording of the conclusions should be used or more definitive data provided

Reply:

We thank the reviewer for the comment. We agree that the phospho-STAT1 signals in IFN λ treated CD4+ and CD8+ cells have the same intensity. However, in CD4+ cells, STAT1 is already phosphorylated in unstimulated cells, and there is no increase of signal intensity upon IFN λ stimulation. We therefore concluded that CD4+Tcells do not respond to IFN λ .

In figure 2C, IFNL4 leads to substantially less IFI27 induction, suggesting that the activity of this protein prep is less active than the preparation of IFN α and IFNL1, this could be caused by technical difficulties in preparing the IFNL4 protein. Nevertheless, the effect of this on the conclusion drawn should be discussed.

Reply:

We thank the reviewer for the comment. Indeed, the effect of IFNL4 on CD69 expression (Figure 2A) and on IFN stimulated gene expression (as suggested by reviewer 2, we have quantified additional ISGs, shown now in revised figure 2) is consistently less strong compared to IFNL1, but this difference is not statistically significant. We have no good explanation for this, because at the level of STAT1 phosphorylation, both IFNL1 and IFNL4 are equally potent (Figure 1D). We therefore prefer not to elaborate on these differences between the IFNLs. The main point of the data shown in figure 2 is that IFNL1 and IFNL4 both behave like "normal" IFNs, i.e. they stimulated B cells.

Minor point.

Please check references carefully, some references seem misplaced below are a few examples
p. 15 first paragraph "The link between IFNL4 genotypes and the cellular immune response against HCV remains to be determined (Sommerreyns et al., 2008). This reference seems to be misplaced, it is used again in the next sentences where it actually appears to belong.

Reply:

We thank the reviewer. We have removed the reference after the sentence "The link between IFNL4 genotypes and the cellular immune response against HCV remains to be determined".

P18 end of page "Clarification of these possibilities would require detailed analysis of intrahepatic lymphocytes isolated from a large cohort of CHC patients because IFN λ 4 gene is not expressed in mice (Key et al., 2014)." The reference to Key et al seems out of context.

Reply:

We thank the reviewer. We have replaced this reference with "Wack A, Terczyńska-Dyla E, Hartmann R. Guarding the frontiers: the biology of type III interferons. *Nat Immunol.* 2015 Aug;16(8):802-9. doi:10.1038/ni.3212. Review. PubMed PMID: 26194286."

Reviewer #2 (Comments to the Authors (Required)):

These authors report that CD19+ B cells and, to a lesser degree, CD8+ T cells express IFN-lambda receptors and respond to treatment with recombinant human IFN-lambda 1 and -lambda 4.

1. The authors should consider examining the effects of lambda 1 and lambda 4 on neutrophils because there are several reports showing that neutrophils are IFN-lambda-responsive.

Reply:

We thank the reviewer for this suggestion. We have tested the response of neutrophils to IFNL4 in two different assays, and provide the results in an additional paragraph in the result section and a new figure (Supplementary Figure 1).

2. The title of this paper ("Interferon lambda 4 directly activates human CD8+ T cells") seems odd in view of the fact that the data provided in Figs. 2 and 3 concern CD19+ B cells instead of CD8+ T cells. I recommend that the authors revise the title as follows: "Interferon lambda 4 can directly activate human CD19+ B cells and CD8+ T cells." Alternatively, if the authors remove Figs. 2 and 3, the current title would be appropriate.

Reply:

We thank the reviewer for this suggestion, and have changed the title accordingly.

3. Although B cells appear to express IFN-lambda receptors, the results in Fig. 2 show that IFN-lambda exerts minimal direct effects on B cell activation as measured by changes in CD69 expression levels.

Reply:

We agree with the reviewer's comment. We explicitly write this now in the discussion (paragraph 4).

4. What is the rationale for choosing IFI27 as the only ISG to confirm IFN responsiveness of B cells in Fig. 2? The authors should measure expression of several, additional, more common ISGs such as MX1, OAS and IFIT1.

Reply:

We thank the reviewer for this suggestion. We have now investigated also the expression of *MX1*, *OAS1* and *IFIT1*. Relevant data are shown in the revised Figure 2 (new panel 2C).

5. Based on their results in Fig. 3, the authors conclude that IFN-alpha and -lambda increase production of IL-10 by CpG-stimulated B cells. In Fig. 4, the authors evaluated the effects of IFN-lambda 1 and 4 on IFN-gamma production by CD8+ T cells. IFN-lambda 1 and 4 weakly enhanced IFN-gamma production, but this effect was only observed when the cells were stimulated with a very low concentration of anti-CD3 mAb. The significance of this observation is unclear.

Reply:

We thank the reviewer for outlining this important aspect. Stimulation of human T cells is often achieved by using anti-CD3 monoclonal antibodies (mAbs). However, stimulation with optimal mAbs doses leads to strong T cell response, which often are different from those observed after antigen stimulation. To obviate this problem, low doses of anti-CD3 mAbs are used, which instead more closely resemble physiological stimulation by antigen. We have now discussed this issue.

T cell receptor activation by anti-CD3 induces expression of multiple cytokines, including IL-10 and IL-22. Therefore, it would be informative to measure the levels of IL-10 and IL-22 produced by T cells after activation with anti-CD3 +/- IFN-alpha and/or IFN-lambda. This would help to broaden the scope of this report.

Reply:

We thank the reviewer for this comment. Using MSD-UPlex we also measured the amounts of released IFN γ , IL10, IL22, IL1 β , TNF α and GM-CSF in CD8+T cells stimulated with suboptimal dose of anti-CD3 in the presence of IFN- α and IFN- λ . These results have been included in revised Figure 2 (panel 2C) and we have added them in the result section "IFNL4 enhances IFN γ production by TCR stimulated CD8 T cells", second paragraph. The new findings are discussed in a additional paragraph (paragraph 7) in the discussion section.

General comment: It is difficult to gauge the biological significance of the findings described in this paper because the magnitude of responses induced by IFN-lambda 1 and 4 (e.g., phospho-STAT1 activation, changes in CD69 expression, or changes in IL-10 expression) are very weak/modest.

Reply:

We agree that IFNLs are weak stimulators (compared to IFN α). However, their effects are important as they synergize with TCR-mediated T cell activation, and thus have relevance during antigen-driven T cell activation. This is in accordance with many studies that report the effect of IFNLs on different cell types. Here, we expand

this observation to human CD8+ T cells. Because the T cell immune response is central to spontaneous clearance of HCV, our findings are biologically significant, at least in the context of HCV infections.

Several previous related studies by others have also examined the ability of various types of blood leukocytes to respond to IFN-lambda. For example:

Freeman J, et al. (2014) Pegylated interferons Lambda-1a and alfa-2a display different gene induction and cytokine and chemokine release profiles in whole blood, human hepatocytes and peripheral blood mononuclear cells. *J Viral Hepat.* 21(6):e1-9.

Yin Z, et al. (2012) Type III IFNs are produced by and stimulate human plasmacytoid dendritic cells. *J Immunol.* 189(6):2735-45.

Kelly A, et al. (2016) Immune cell profiling of IFN-λ response shows pDCs express highest level of IFN-λR1 and are directly responsive via the JAK-STAT Pathway. *J Interferon Cytokine Res.* 36(12):671-680.

Reply:

We thank the reviewer for this comment that stimulated us to read these papers again. We conclude that these interesting papers report the following key findings:

The study of Freeman J, *et al.* (2014) described the response of PBMCs to IFNL stimulation without identifying the responsive population in PBMCs.

The study of Yin Z, *et al* (2012) describe plasmacytoid dendritic cells as source of IFNL and also responsive population.

The study of Kelly A, *et al* (2016) identified plasmacytoid dendritic cell as responsive population. The study also showed that monocytes and B cells expressed IFNLR1.

We think that our paper is the first to report that IFNL1 and IFNL4 directly bind to human B and CD8+ T cells and have a direct impact on them.

October 12, 2020

RE: Life Science Alliance Manuscript #LSA-2019-00612-TR

Prof. Markus H Heim
Basel, University
Department of Research
Universitaetsspital Basel
Hebelstrasse 20
Basel, BS 4031
Switzerland

Dear Dr. Heim,

Thank you for submitting your revised manuscript entitled "Interferon lambda 4 can directly activate human CD19+ B cells and CD8+ T cells". We would be happy to publish your paper in Life Science Alliance pending final revisions necessary to meet our formatting guidelines, and minor suggestions from reviewers.

Along with the points listed below, please also attend to the following:

- please make sure that the author names in the manuscript and the names in our system match
- please have secondary corresponding author add their ORCID ID-they should have received instructions on how to do so
- please use the [10 author names, et al.] format in your references (i.e. limit the author names to the first 10)
- please add a callout for Figure 4, Panels I & J and Figure 5, Panels B&C in the main manuscript text

A. FINAL FILES:

-- High-resolution figure, supplementary figure and video files uploaded as individual files: See our detailed guidelines for preparing your production-ready images, <https://www.life-science->

alliance.org/authors

B. MANUSCRIPT ORGANIZATION AND FORMATTING:

Sincerely,

Shachi Bhatt, Ph.D.
Executive Editor
Life Science Alliance
<https://www.lsjournal.org/>
Tweet @SciBhatt @LSAJournal

Reviewer #1 (Comments to the Authors (Required)):

This is a revised version: The authors has addressed my concerns and I recommend publication of this revised manuscript.

Reviewer #2 (Comments to the Authors (Required)):

I have reviewed the point-by-point responses from the authors regarding the criticisms I raised during my review of the original version of this paper. The authors have satisfactorily addressed my review comments, and in my opinion, this revised manuscript is now acceptable for publication. However, I do have one minor text change (see below) that I think the authors should make before this paper is accepted.

In the Abstract, this sentence is too long and difficult to understand: "Multidimensional flow cytometry of cells from liver biopsies of hepatitis patients showed accumulation in IFNL4-producers of CD8+ T cells with central memory-like, partial activation and not senescent/exhausted phenotype, which instead were more abundant in IFNL4-non-producers." Please split this sentence into two shorter sentences as follows:

"Multidimensional flow cytometry of cells from liver biopsies of hepatitis patients from IFNL4-producers showed accumulation of activated CD8+ T cells with a central memory-like phenotype. In contrast, CD8+ T cells with a senescent/exhausted phenotype were more abundant in IFNL4-non-producers."

This text revision will make it easier for readers to understand the point the authors are trying to communicate here.

October 26, 2020

RE: Life Science Alliance Manuscript #LSA-2019-00612-TRR

Prof. Markus H Heim
Basel, University
Department of Research
Universitaetsspital Basel
Hebelstrasse 20
Basel, BS 4031
Switzerland

Dear Dr. Heim,

Thank you for submitting your Research Article entitled "Interferon lambda 4 can directly activate human CD19+ B cells and CD8+ T cells". It is a pleasure to let you know that your manuscript is now accepted for publication in Life Science Alliance. Congratulations on this interesting work.

*****IMPORTANT:** If you will be unreachable at any time, please provide us with the email address of an alternate author. Failure to respond to routine queries may lead to unavoidable delays in publication.*******

DISTRIBUTION OF MATERIALS:

Again, congratulations on a very nice paper. I hope you found the review process to be constructive and are pleased with how the manuscript was handled editorially. We look forward to future exciting submissions from your lab.

Sincerely,

Shachi Bhatt, Ph.D.

Executive Editor

Life Science Alliance

<https://www.lsjournal.org/>
